# Best Practices in Pool-based Active Learning for Image Classification

## Abstract

The recent popularity of active learning (AL) methods for image classification using deep-learning has led to a large number of publications that contributed to significant progress in the field. Benchmarking the latest works in an exhaustive and unified way and evaluating the improvements made by the novel methods is of key importance to advance the research in AL. Reproducing state-of-the-art AL methods is often cumbersome, since the results and the ranking order of different strategies are highly dependent on several factors, such as training settings, used data type, network architectures, loss function and more. With our work we highlight the main factors that should be considered when proposing new AL strategies. In addition, we provide solid benchmarks to compare new with existing methods. We therefore conduct a comprehensive study on the influence of these key aspects, providing best practices in pool-based AL for image classification. We emphasize aspects such as the importance of using data augmentation, the need of separating the contribution of a classification network and the acquisition strategy to the overall performance, the advantages that a proper initialization of the network can bring to AL. Moreover, we make a new codebase available, that enables state-of-the-art performance for the investigated methods, which we hope will serve the AL community as a new starting point when proposing new AL strategies.

## 1 Introduction

Deep learning methods require large amounts of labeled data samples to train. Unfortunately, annotating new datasets consisting of thousands or millions of images is very costly. A research topic that focuses on maximizing the performance of deep learning models with a given annotation budget is active learning (AL). AL is a machine learning's sub-field in which the algorithm is allowed to query an information source for the label of new data samples (Settles, 2009). Recently, an abundance of pool-based AL methods for image classification have been proposed (Caramalau et al., 2021a;b; Kim et al., 2021; Liu et al., 2021; Choi et al., 2021). In pool-based AL (Lewis & Gale, 1994) there exists a large unlabeled $\mathcal{U}$ and a small labeled sample set $\mathcal{L}$. The labeled set is used to train the classifier and the large unlabeled set is used to query new samples that then are annotated and added to the labeled set in order to include them for training. The goal of pool-based AL is to iteratively sample and annotate unlabeled training samples from the pool and add them to the training set in order to achieve the best performance given a certain annotation budget. Pool based AL is often particularly useful because large amounts of unlabeled data is often available but annotating it is cumbersome and costly.

In AL there exist multiple baselines that either form lower or upper bounds for AL methods. Typically, using all the unlabeled samples, annotate them and include them for training forms an upper bound that AL methods aim to reach with as few samples as possible. In contrast, selecting new samples that should be labeled and used for training randomly from the set of unlabeled samples establishes a lower bound that any AL method should exceed in every AL cycle. AL methods can be split into uncertainty or diversity based samples. The former select samples for which the classifier is uncertain (Gal et al., 2017; Beluch et al., 2018; Yoo & Kweon, 2019; Mayer & Timofte, 2020), while the latter selects samples in a diverse way (Sener & Savarese, 2018; Sinha et al., 2019; Gissin & Shalev-Shwartz, 2019; Caramalau et al., 2021a), ensuring to cover the whole data distribution of the unlabeled set. Recently proposed strategies, are often still based on these concepts, combine

them (Ash et al., 2019; Zhang et al., 2020; Kim et al., 2021) or propose new ideas such as using unlabeled data to train addition modules (Shui et al., 2020; Caramalau et al., 2021b) used for sample selection or combine AL with semi-supervised learning (Sim'eoni et al., 2021; Huang et al., 2021; Gao et al., 2020; Mittal et al., 2019; Liao et al., 2021). The recent popularity of AL methods complicates comparing these works in a complete and standardized manner. However, assessing the improvements made by new methods is crucial to boost the progress in AL. Hence, this work provides best practices to keep in mind, when proposing new pool-based AL methods for image classification using deep learning and comparing them to existing methods form the literature.

**Contributions:** In summary, our contributions are as follows: **(i)** We perform an extensive analysis of popular AL methods and study the effect of different settings typically used in the literature. **(ii)** In addition, we discuss the merit of recent trends in AL such as using unlabeled data, pretraining or initial set construction. **(iii)** We incorporate our findings into a collection of best practices to keep in mind when evaluating pool based AL methods for image classification. **(iv)** We make a new Pytorch code base available [1] for AL that contains many popular AL methods and datasets and allows to evaluate and compare different strategies in a fair way. Code will be released upon publication.

## 2    RELATED WORK

**AL evaluations:** Recently the attention on benchmarking and unifying results of new AL strategies increased. Beck et al. (2021) conducted a rich set of experiments to evaluate the performance of AL methods, including (Wei et al., 2015; Sener & Savarese, 2018; Ash et al., 2019; Killamsetty et al., 2021). They assessed the robustness of the methods by adding redundancy to the datasets, concluding that diversity based strategies are more stable in this scenario. In addition, they studied the effect of using data augmentation, using different optimizers, updating the models instead of re-initializing them in every iteration, and using different batch sizes. In contrast, we perform similar experiments but include more AL methods on multiple datasets, evaluated additional settings, and study the effect of using unlabeled samples for AL. Munjal et al. (2020) focused on benchmarking the strategies proposed in (Gal et al., 2017; Beluch et al., 2018; Sener & Savarese, 2018; Sinha et al., 2019) using a uniform experimental setting. They vary the AL batch size, the validation set size, and the amount of class imbalance. They find that under changing experimental condition no strategy offers gain over Random sampling. Conversely, our experiments show that AL methods clearly outperform Random sampling when training the classifier carefully. Chong et al. (2021) evaluates traditional AL methods and Coreset on an imbalanced medical dataset (Rahman et al., 2021) and on CIFAR10. They highlight the importance of respecting class imbalance in AL and conclude that Coreset performs better than Uncertainty sampling. In contrast to these last two works, we omit analyzing the robustness of AL strategies on imbalanced datasets but rather focus on balanced datasets such as CIFAR10.

**Investigated AL strategies:** The investigated AL strategies are a relevant selection of recent deep pool based AL methods used in deep learning. In particular, we study Coreset, BADGE, LL4AL, JLS and WAAL, and Uncertainty sampling. We briefly summarize the most recent ones. Coreset (Sener & Savarese, 2018) selects new unlabeled samples by solving the k-Center problem (Wolf, 2011) on the feature space of the last fully-connected layer. Ash et al. (2019) propose an acquisition strategy (BADGE) based on k-MEANS++ (Arthur & Vassilvitskii, 2007), which is applied on so called hallucinated gradients with respect to the parameters of the final layer of the deep network. Hallucinated gradients are used because the true labels are not available for unlabeled samples. Therefore, the most likely labels according to the classifier are used to compute the gradient. Yoo & Kweon (2019) propose LL4AL, based on the assumption that data points for which the classification network is likely to produce a wrong prediction are most informative and should be used for training. To this end, they jointly train a target module and a *loss prediction module* that aims at predicting the loss of a given unlabeled sample. The samples achieving the highest losses are then added to the training set. JLS (Caramalau et al., 2021b) employs a discriminator network besides the classifier. The task of the discriminator is differentiating whether a given sample belongs to the set of labeled or unlabeled samples. The samples that the discriminator predicts to be unlabeled with a high confidence are selected and annotated. JLS shares intermediate backbone layers for classification and discrimination such that training the discriminator with unlabeled data turns out to be beneficial for

---

[1]The code will be made available after publication.

the classifier as well. Shui et al. (2020) propose an acquisition strategy (WAAL) that combines the uncertainty score of the task model with a diversity term based on the Wasserstein metric. To compute the diversity metric, an adversarial network is used that is optimized to discriminate between labeled and unlabeled data. Both networks share the same feature extractor and are jointly trained through a min-max optimization problem that uses labeled and unlabeled data.

## 3   EXPERIMENTAL SETUP

Since the accuracy range of AL methods extensively changes across the literature and their provided codebases, we provide our own that aims at maximizing AL accuracy for all the implemented methods. Our codebase builds upon a repository[2] which only implements the LL4AL strategy for CIFAR10 and achieves the result presented in (Yoo & Kweon, 2019). We re-implement multiple other methods such as Random sampling, Uncertainty sampling, Coreset (greedy K-center variant), BADGE, JLS, VAAL and include them into the codebase. Furthermore, we adapt the codebase in order to support additional datasets such as SVHN and FashionMNIST. The different methods can be tested on any dataset and our codebase allows to examine the effect of various different settings such as warm or cold start, different classifiers, unsupervised pretraining, changing the acquisition strategy to sample the initial dataset, different training settings, such as jointly training a discriminator or a loss prediction module for all methods.

We use the settings in (Yoo & Kweon, 2019) because they achieve results with high accuracy on CIFAR10 for all the reported methods. Note that the default settings serves mainly as comparison benchmark, and does not lead to the highest absolute accuracy for each method across all datasets. Unless stated otherwise, our default training setting is as follows: We use ResNet-18 (He et al., 2016) as backbone and train with a mini-batch size of 128, for 200 epochs, with SGD optimizer, an initial learning rate of 0.1, momentum of 0.9 and weight decay of 0.0005. After training for 160 epochs, we decrease the learning rate to 0.01. Instead of initializing the network weights in every AL cycle we use the weights of the network obtained in the previous AL cycle (warm start). We use data augmentation (random horizontal flips and random crop) and normalize the input images. The initial training set consists of 1000 labeled samples. We add 1000 samples in each AL step, except for the experiments focusing on the use of unlabeled data where we use the JLS and WAAL settings.

Our codebase allows to reproduce state-of-the-art results for the aforementioned methods for all three datasets, for different training setting and annotation budgets. All results reported in the paper are averaged over three runs with different random seeds.

For the experiments we use three common datasets used in the AL literature: **CIFAR10** (Krizhevsky, 2009), consisting of 60k $32 \times 32$ RGB images, divided into 50k training and 10k testing images split into 10 classes. The datasets are balanced, i.e. each class has the same number of images. **SVHN** (Netzer et al., 2011) contains images showing house numbers obtained from Google Street View. It consists of 73,257 training, 26,032 testing, and 53,1131 extra training images. The RGB images have a resolution of $32 \times 32$, are centered around a single digit and are split into 10 classes. The extra training images are not used in this work. **FashionMNIST** (Xiao et al., 2017) contains $28 \times 28$ gray-scale images showing articles from an online shop. The dataset consists of 60k training and 10k testing images.

Figure 1 shows the results when using our codebase and the default settings on three datasets. In the paper we often report only the accuracy difference plots but omit the figures showing the absolute accuracy, or we only show the result on a single dataset, or we omit other AL strategies such as VAAL or JLS to keep the figures clean, or we only report the mean but omit the standard deviation. For additional plots containing all these results we refer the reader to the Appendix.

## 4   DIFFERENT TRAINING SETTINGS AND THEIR EFFECT ON AL EVALUATION

Newly proposed AL strategies typically come with their own hyper parameters, training strategies and model architectures in order to outperform existing methods on the same dataset. However, changing these aspects complicates comparing different AL methods with each other. For example, when using another classification network (backbone) than existing methods a fair comparison

---

[2]https://github.com/Mephisto405/Learning-Loss-for-Active-Learning

Table 1: Comparison of existing AL methods in terms of model architecture, optimizer, learning rate, whether data augmentation is used or not and whether cold or warm start is used to initialize the network parameters. '?'- means that the information was not mentioned.

| | Coreset Sener & Savarese 2018 | VAAL Sinha et al. 2019 | BADGE Ash et al. 2019 | LL4AL Yoo & Kweon 2019 | JLS Caramalau et al. 2021b | WAAL Shui et al. 2020 | SRAAL Zhang et al. 2020 | TA-VAAL Kim et al. 2021 | ISAL Liu et al. 2021 | FKSSAL Gudovskiy et al. 2020 | GCN Caramalau et al. 2021a | VaB-AL Choi et al. 2021 |
|---|---|---|---|---|---|---|---|---|---|---|---|---|
| Backbone | VGG-16 | VGG-16 | ResNet-18 | ResNet-18 | VGG-16 | VGG-16 | ResNet-18 | ResNet-18 | ResNet-18 | ResNet-18 | ResNet-18 | ResNet18 |
| Optimizer | RMSProp | SGD | Adam | SGD | SGD | SGD | ? | SGD | SGD | SGD | SGD | SGD |
| LR | 0.001 | 0.01 | 0.01 | 0.1 | 0.01 | 0.01 | ? | 0.1 | 0.1 | 0.1 | 0.1 | 0.1 |
| Data Aug | ? | ✓ | ✗ | ✓ | ✓ | ✓ | ? | ✓ | ✓ | ✗ | ✓ | ? |
| Start | Cold | Cold | Cold | Warm | Cold | Cold | Warm | Cold | Cold | Cold | Cold | ? |

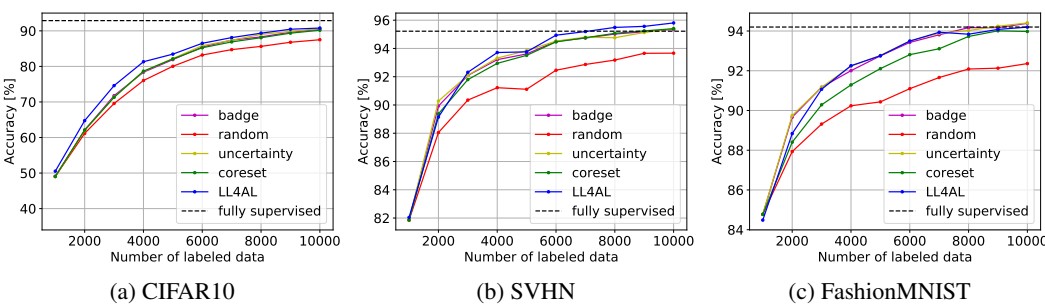

| (a) CIFAR10 | (b) SVHN | (c) FashionMNIST |
|---|---|---|

Figure 1: Results for the default setting of the AL strategies using our codebase.

requires to re-implement these methods or change the provided source code. However, the default hyper parameters and training settings may not be optimal for the new backbone. In Table 1 we summarize settings for recent AL methods. We observe multiple choices for all the reported settings. In order to enable a fair comparison among AL methods a unified evaluation strategy is required. Hence, we examine in this section the effect on AL when using the different settings by changing the default values in our proposed code base. In particular, we study whether changing the settings affect only the overall accuracy, leads to a change in the ranking of AL methods or even results into a vanishing benefit of AL. We compare four existing AL methods (LL4AL, BADGE, Uncertainty sampling and Coreset) with each other and the baseline of Random sampling. Finally, we compare our code base with the code base used for BADGE that not only provides the implementation of BADGE but also of a few existing methods.

**Backbone Architecture:** First, we investigate the impact on AL performance when changing the backbone. In particular, we compare five methods on CIFAR10 using either VGG-16, ResNet-18 or DenseNet-201. Figure 2 shows the achieved accuracy for the different backbones. We conclude, that different backbones achieve different classification accuracy, with ResNet-18 achieving the highest. Hence, it is crucial to ensure that only methods using the same backbone are compared with each other. In addition, we observe that the ranking order of AL methods changes for Densenet-201 compared to ResNet-18 and VGG-16. This probably due to overall low accuracy leading to poor sample selection. Therefore, we propose to validate new methods on ResNet-18 since it achieves the highest accuracy and significant improvements over Random sampling for all investigated methods. For methods that benefit from using other backbones we recommend to validate for this case on both the new backbone and ResNet-18 to enable a fair and insightful comparison.

**Initializing Backbone Weights:** In AL the classifier is re-trained in every active learning cycle to leverage the newly sampled and annotated data. According to Table 1 two different strategies are common in the literature: cold and warm start. Cold start describes the approach to re-initialize the network weights in every cycle from scratch. In contrast, warm start uses the network weights obtained in the former AL cycle by training with the corresponding annotated samples. Warm start resembles fine-tuning but we use the same learning rate and number of training epochs for cold and warm start. Figure 3 shows the accuracy achieved with **warm** start **minus** the accuracy of **cold** start on three datasets. We observe that depending on the dataset either cold or warm start performs better in the first few AL cycles but the difference diminishes on all datasets for almost all methods. Except LL4AL that performs better with warm start across all datasets. Figure 4c compares the performance of LL4AL with other AL methods when all use cold start and we observe that LL4AL no longer outperforms other AL methods (see figure 1a) but achieves comparable accuracy.

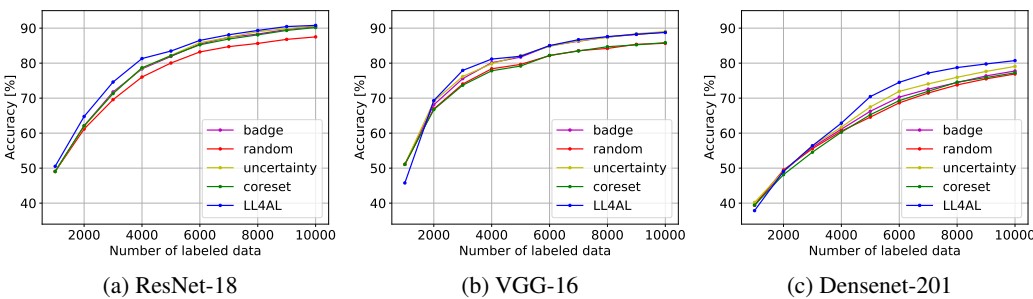

Figure 2: Results when employing different backbones as classifiers.

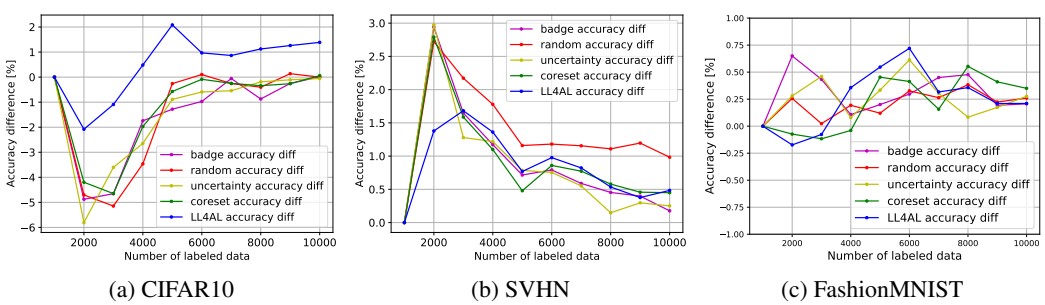

Figure 3: Accuracy difference between Warm-start and Cold-start.

Hence, we conclude that cold or warm start influences AL performance in particular at the beginning of AL but the difference tends to diminish later on. To ensure that the improvement of an AL methods originate from a superior sample selection strategy instead of from improved settings method, we advise to use the approach which works best for the investigated method. For LL4AL we advise to always use warm start, while for the other methods it depends on the annotation budget and on the dataset.

**Optimizer and Learning Rates:** Different optimizers such as SGD, Adam and RMSProp with varying learning rates from 0.001 to 0.1 are commonly used in the literature (see Table 1). First, we investigate the effect of varying learning rates when using the most popular optimizer SGD. We perform experiments when using cold and warm start to initialize the network weights, motivated by the assumption that different initialization approaches may require different learning rates. Figures 4a and 4b show the accuracy achieved with a learning rate of **0.1 minus** the accuracy with **0.01** for CIFAR10. We observe that using a larger learning rate is beneficial for both, cold and warm start. In addition, a smaller learning rate leads to higher classification accuracy when training only on the initial dataset.

Furthermore, we perform experiments with two different optimizers: SGD and Adam. We tested different learning rates used in the literature (see Table 1) and use the learning rate 0.1 that leads to the best performance for either optimizer (see A.13 for more details). Figure 5 shows the accuracy achieved with **SGD minus** the accuracy achieve when using **Adam**. We observe SGD leads in particular on CIFAR10 to superior performance by achieving roughly 6% points higher accuracy than training with Adam. On SVHN and FashionMNIST SGD outperforms Adam as well but the difference is smaller with 0.5 - 1.0% or 0.5 - 1.5% points.

Hence, we conclude that using the same optimizer and learning rate to train the backbone when comparing different AL methods is crucial. Otherwise it is unclear whether the performance difference stems from the AL method or the training setup.

**With and Without Data Augmentation:** Whether to train the backbone with or without data augmentation is another setting that often differs between different AL works (see Table 1). Disabling data-augmentation typically decreases the overall accuracy of the classifier (see figure 7) and can affect the ranking order of AL methods. Figure 6a shows the result on our code base with and without

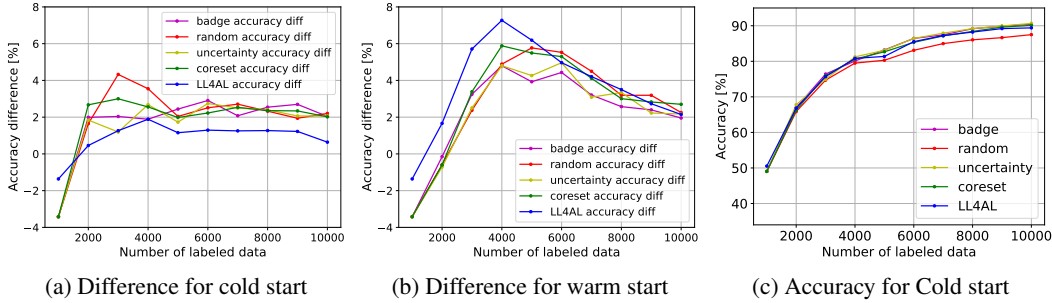

Figure 4: Accuracy difference between $lr = 0.1$ and $lr = 0.01$ with Warm-start, and Cold-start. Results on CIFAR10 when using a Cold-start approach and the default learning rate of 0.1.

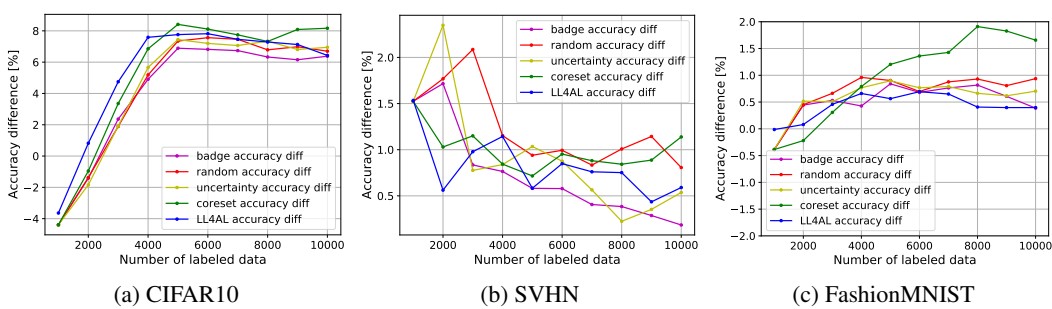

Figure 5: Accuracy difference when using either SGD or Adam as optimizer.

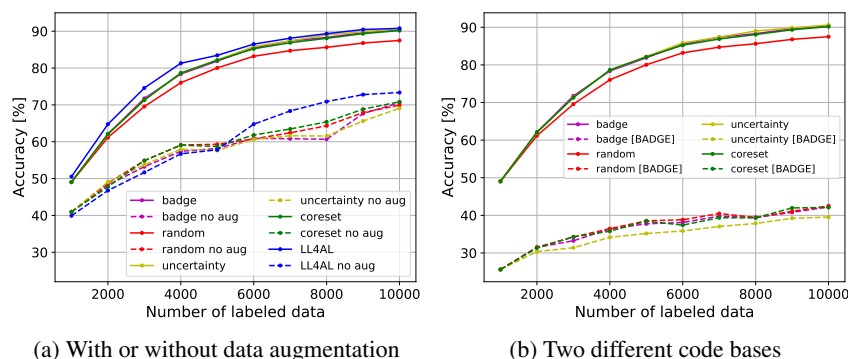

Figure 6: Results on CIFAR10 for different code bases or *with* and *without* data augmentation.

data augmentation on CIFAR10. We observe, that the accuracy is at least 10% points lower when not using data augmentation and can differ up to 20% points and more. In addition, all AL methods outperform random sampling when using data augmentation. Conversely, the baseline random sampling outperforms all AL methods at least once on different operating points when data augmentation is not used. Hence, disabling data augmentation not only decrease the overall performance but also leads to failure of multiple AL methods.

Supported by the above observations we therefore recommend data augmentation when comparing AL for image classification.

**Comparing Codebases:** Having examined all the aforementioned settings for AL it becomes clear that these settings are crucial when comparing AL methods. While it is important to enable a fair environment when comparing different methods, we argue that it is useful to use a codebase that aims at maximizing the overall accuracy. In this part, we want to highlight the importance of such a codebase by comparing it to the codebase used for BADGE. In contrast to our codebase they

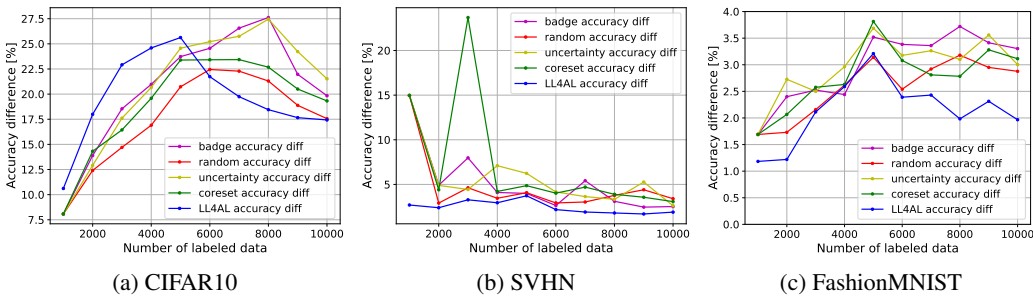

| (a) CIFAR10 | (b) SVHN | (c) FashionMNIST |

Figure 7: Accuracy difference with data augmentation enabled or disabled.

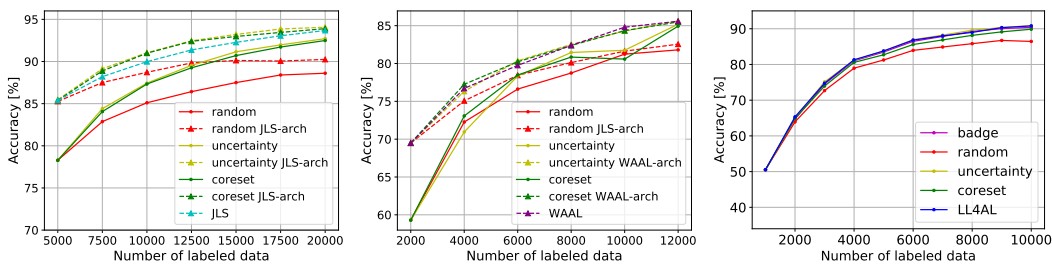

(a) With and without JLS Arch.  (b) With and without WAAL Arch.  (c) With the loss prediction module

Figure 8: Results when jointly training the discriminators for different methods and their performance without training it or when jointly training the loss prediction module for all methods.

use Adam instead of SGD, disable data augmentation and train for fewer epochs in each AL cycle and employ early stopping. Figure 6b shows the results of four AL methods using our codebase in solid lines and the results of BADGE's codebase in dashed lines. We observe that our codebase leads to much higher accuracy. In particular we achieve between 20% to almost 50% points higher classification accuracy on CIFAR10 using the same annotation budget. Furthermore, we observe that every AL method outperforms Random sampling when using our codebase. In contrast, the baseline often outperforms AL methods when using the codebase of BADGE and therefore results in AL failure. However, this is not a problem of the AL methods but is caused by the codebase. We conclude that poorly trained backbones perform poorly for sample selection and therefore result in low performance of the AL strategy. Furthermore, it is clear that a codebase achieving too low classification accuracy is unsuitable for a fair comparison of different AL methods.

# 5 USING UNLABELED DATA

In this section we focus on the use of unlabeled data. Recent AL methods such as VAAL, LL4AL or JLS employ an additional module that is trained in each active learning cycle and used to query new samples. While most AL methods do not use unlabeled data to train the classification network (e.g., VAAL), or only use labeled data (e.g., LL4AL), there exist methods (e.g., JLS, WAAL) that use unlabeled data to train the backbone. Hence, comparing methods that use unlabeled samples with methods that only use labeled samples is unfair. In order to compare such methods with existing ones, we propose to train the backbone of all methods with unlabeled samples. We perform this experiments using the training strategies of JLS and WAAL. In particular, we use exactly the same network for each AL strategy including the discriminators but we use the original sample selection strategy when querying new samples in each AL cycle. Figure 8a shows a comparison between the performance of AL methods when training the classifier and discriminator with unlabeled data (dashed lines) compared to their performance when only the backbone is trained (solid lines). We observe that traditional methods such as Coreset and Uncertainty sampling outperform JLS when using the same networks and train with unlabeled samples. Figure 8b shows similar results for WAAL using their codebase instead of ours.

## 6 EFFECT OF ADDING MODULES FOR COMPLEMENTARY TASKS

In this section, we demonstrate, that it is important to separate the achieved improvement caused by the acquisition strategy and a better trained classifier when learning additional tasks beside classification. LL4AL trains the backbone combined with the *loss prediction module*, optimizing thus a different loss function. To fairly asses the benefit of their acquisition strategy, we train each AL method using exactly the same architecture as in LL4AL, but use the original sampling strategy when querying new images. In figure 8c all methods train jointly also the *loss prediction module*. Comparing the results to the default case, depicted in figure 1a, we observe that LL4AL is not achieving the top performance anymore. The other methods achieve comparable accuracy. This experiment shows that it is not the novel acquisition strategy proposed in LL4AL that leads to superior performance, but it is in fact the classifier that benefits of learning an additional task and results in better generalization.

## 7 BEFORE ACTIVE LEARNING STARTS

In this section we discuss two approaches that can influence the performance of AL methods, by either building the initial labeled dataset following a certain strategy instead of random sampling of the data or by pretraining the backbone in an unsupervised setting.

**Construction of the Initial Training Set:** Instead of randomly selecting the samples for the initial training set, it is possible to use a more sophisticate approach to construct the dataset. In this part, we investigate whether it is worth focusing on selecting such a set and if a potential benefit is preserved over multiple AL cycles or diminishes quickly. In particular, we use a pretrained VGG-16 on ImageNet (Deng et al., 2009) and compute the features of its last layer for every image in the datasets. We then use K-means++ and greedy K-center on the extracted features to select the set of initial training samples. We evaluate different AL methods and report in figure 9 the accuracy of **randomly** created initial sets **minus** the accuracy achieved when using **K-means++/K-center** to create the initial training set.

We observe in figure 9a that in the first iteration the accuracy improves for all methods beside LL4AL using K-means++ based initialization on CIFAR10 (more than $2\%$ points comparing with the absolute numbers) and that the benefit rapidly vanishes within the next few AL iterations. Imagenet and CIFAR10 both contain natural images in contrast to SVHN that contains images less similar to Imagenet. Therefore, it is of interest to verify if there is still a benefit of another method to create the initial set than random sampling on SVHN. The results depicted in figure 9b show that the samples selected with Kmeans++ result in lower performance in the first iteration. In figure 9c we observe that using K-center for constructing the initial labeled set, deteriorates the accuracy by more than $10\%$ points in the first AL cycle. We assume that a diversity based method probably mainly selects outliers present in the dataset and results therefore in such a poor performance. In contrast, Zhang et al. (2020) experienced a $2\%$ points accuracy improvement on CIFAR10 with 5000 training samples. However, they used the deep latent features of a trained VAE, which encourages dataset specific features that are spread over the latent space according to a defined distribution. One key observation is that the benefit or the drawback of using another method than Random sampling to construct the initial set disappears rapidly within the next few AL cycles.

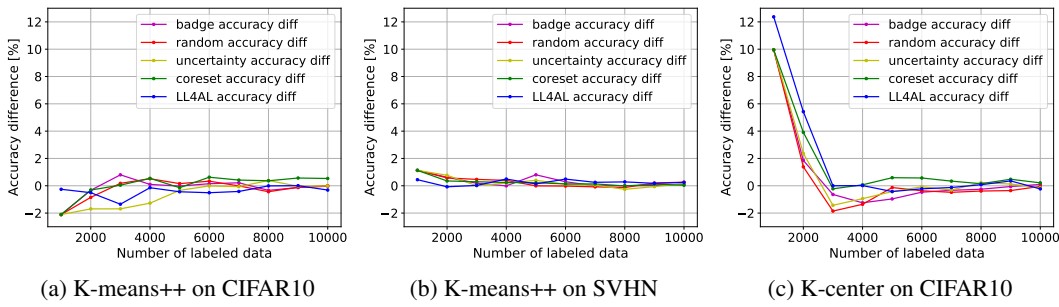

(a) K-means++ on CIFAR10      (b) K-means++ on SVHN      (c) K-center on CIFAR10

Figure 9: Accuracy difference between random initialization and two initialization methods.

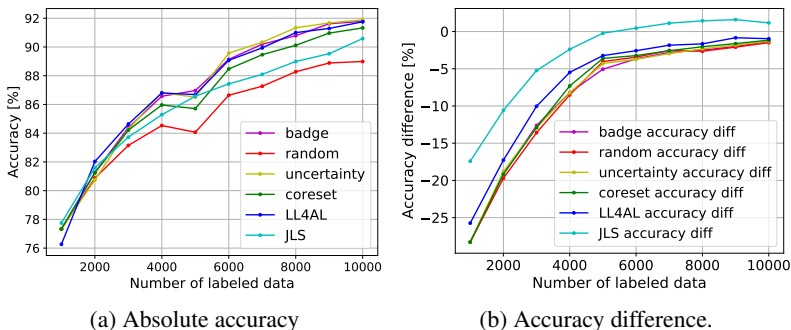

(a) Absolute accuracy          (b) Accuracy difference.

Figure 10: Results on CIFAR10 when using unsupervised pretrained weight at each AL cycle. The accuracy difference corresponds to randomly initialized vs pretrained weights.

**Unsupervised pretraining of the network:** Typically, the parameters of the classifier are randomly initialized before AL starts. Conversely, in a warm start scenario, the weights obtained in the previous AL cycle are used in the current instead of re-initializing the weights from scratch. In such a setting, AL performance could be increased by using unsupervised pretraining of the classifier and use these dataset specific features as a starting point for AL. Gidaris et al. (2018) showed a massive improvement in accuracy for the task of classification when using their proposed unsupervised pretraining method. The idea is to train the network on predicting which 2d rotation has been applied to the input image. This initialization method has already been used in AL context by Gudovskiy et al. (2020). Preliminary experiments show that re-initializing the model at every AL cycle with the unsupervised weights compared to using them only for the first iteration is beneficial, thus we followed the former approach.

Figure 10a shows the absolute result when using unsupervised pretraining for different AL methods. In contrast, Figure 10b shows the accuracy of the classifier using **randomly** initialized weights **minus** achieve accuracy when using the **unsupervised** weights. The results show that with the unsupervised pretrained network the overall accuracy improves by $25\%$ in the first AL cycle, and that the accuracy difference diminishes as the number of labeled samples increases. Note, that LL4AL does not outperform the other benchmark in this case, probably because learning the additional task of the loss module does not help as much when the network is already properly trained (compare with the findings in section 6). Furthermore, JLS that in addition to the unsupervised pretrained weights trains with unlabelled data at each training cycle achieves even lower performance than other AL methods. The benefit of jointly training the backbone and the discriminator by leveraging unlabeled data is only visible in the first iteration, but standard acquisition strategies, such as Uncertainty sampling outperform JLS afterwards.

## 8 CONCLUSION

In this paper we provide a comprehensive evaluation of recent deep AL strategies, setting a solid benchmark for different datasets. We analyze different training settings and observe that they can cause overall accuracy differences and ranking shifts between AL methods. Data augmentation and SGD combined with a learning rate of 0.1 provide higher accuracy for all AL methods when using a ResNet-18. We study the effect of different initialization methods in the AL context. In particular, advanced initial set construction methods rapidly reach the performance of a randomly selected one. Furthermore, we observe that using unsupervised pretraining of the network greatly benefits the classification performance especially when the number of labeled samples is small. Future AL methods should try to clearly disentangle the performance improvement obtained by changes in the classification network and by the proposed acquisition strategy. When evaluating a new method, it should ensured that the existing state-of-the-art AL strategies used as benchmarks achieve comparable results with the original accuracy reported on a specific dataset and AL interval. Deviating from the standard settings for a novel method complicates a fair comparison. Nonetheless, in such cases a new method should only compare to the baselines that use the novel settings instead of the standard ones if they achieve better results.

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

## A  APPENDIX

### A.1  DATASETS

The AL strategy must be robust to changes of the dataset.

Figure 11 shows the accuracy that the investigated AL methods have on CIFAR10, CIFAR100, SVHN, FashionMNIST and TinyImageNet.

Overall the AL strategies show an improvement over random sampling across the datasets, beside for TinyImageNet. As we have seen, often AL fails when dealing with poorly trained networks. The overall small accuracy on TinyImageNet and the big variance across the different random seeds indicate that the network is poorly trained and there is still room for further improvements for new AL strategies and training settings.

Table 2 shows the performance of the default configuration when trained on the full dataset.

Table 2: Results on the full datasets

| Dataset | Accuracy [%] |
|---------|--------------|
| CIFAR10 | $92.88 \pm 0.23$ |
| SVHN | $95.22 \pm 0.46$ |
| FashionMNIST | $94.20 \pm 0.17$ |
| CIFAR100 | $73.31 \pm 0.28$ |
| TinyImageNet | $24.56 \pm 0.98$ |

### A.2  BACKBONE

The influence of the backbone used for the classification task is another key aspect that must be taken into account. Since for LL4AL the loss module is trained jointly with the backbone by sharing the features of intermediate layers and no guideline was provided for which to use on different architecture, we chose to use for VGG networks (Simonyan & Zisserman, 2015) the features of the last four max pooling layers and for DenseNet architectures (Huang et al., 2017) the last four transition layers. The influence of different backbones is depicted in figure 12.

Badge, Uncertainty and Coreset performer similarly to Random sampling with Densenet201. ResNet-18 achieves the overall best performance.

### A.3  CODEBASE

The results for different codebases with their standard deviation are depicted in figure 13

### A.4  DATA AUGMENTATION VS NO AUGMENTATION

For all datasets LL4AL has a better relative ranking compared to the other methods when no data augmentation is used. This probably due to the fact that without augmentation the networks are

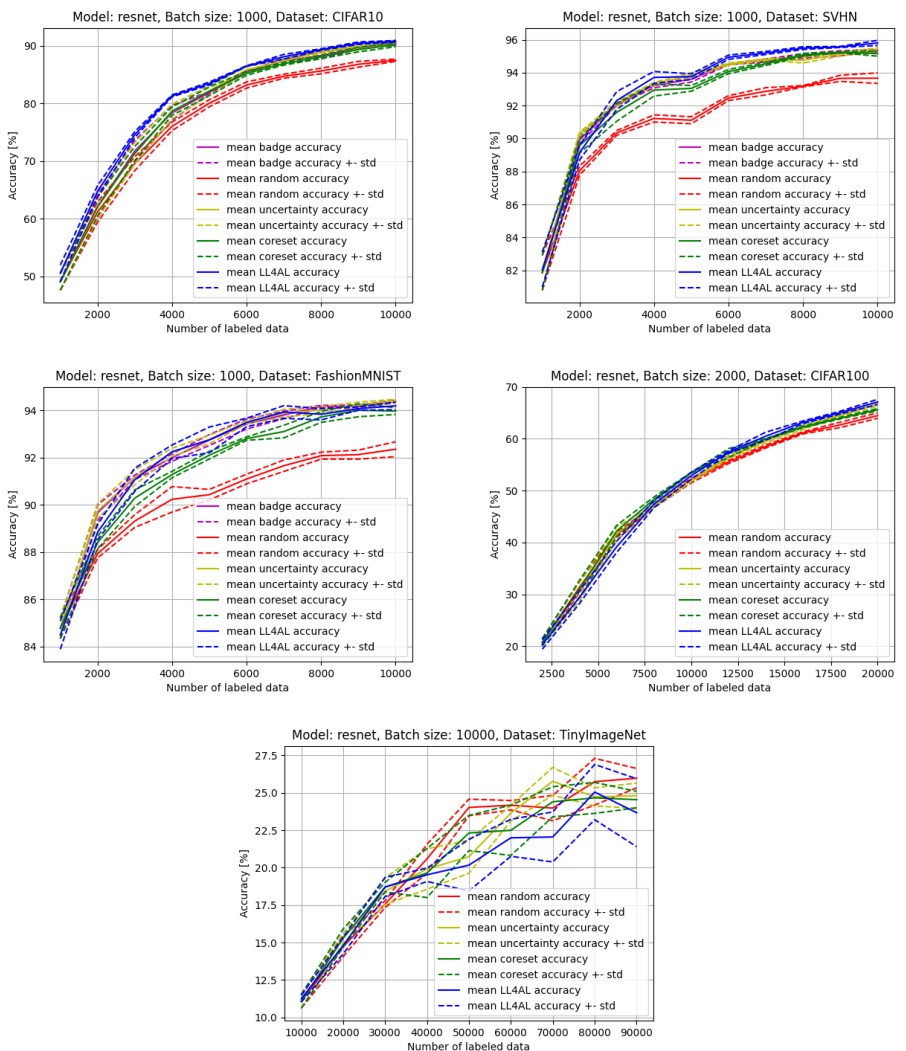

Figure 11: Results for different datasets

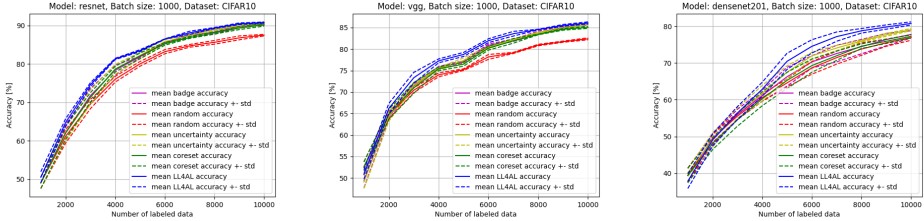

Figure 12: Results for different backbones: ResNet-18 (left), VGG-11 (center), DenseNet-201 (right)

trained worse and the additional task of learning the losses increases more the ability of the classifier to generalize. Overall data augmentation improves significantly the accuracy of AL strategies as can be seen from figure 15.

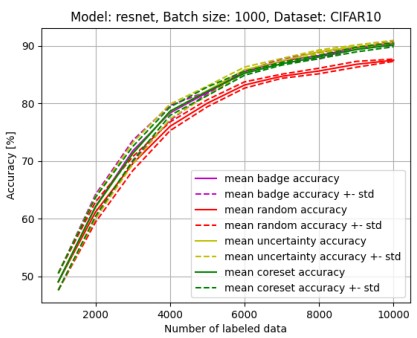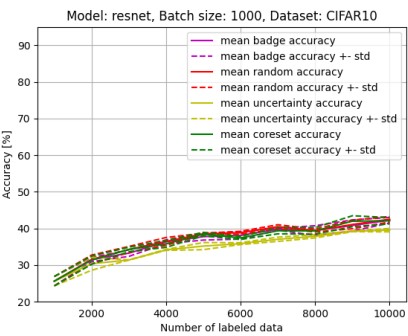

Figure 13: Results on our codebase and on BADGE's codebase

## A.5 OPTIMIZER: SGD VS ADAM

Badge performs similarly to better than LL4AL with the Adam optimizer. The overall accuracy is worse. Coreset performs poorer on the FashionMNIST dataset with Adam. Figure 17 shows the accuracy difference between the default settings using SGD optimizer and the results using Adam optimizer for different datasets.

## A.6 LEARNING RATES

The results of using different learning rates ($LR$) are shown in figure 18.

Using a smaller $LR$ seems beneficial in the first iterations but it worsens the performance as the iterations progress. Since a smaller $LR$ combined with a Warm-start approach may lead to a model that doesn't learn enough from the new added samples, we investigated the influence of different learning rates also with a cold-start approach as can be seen in figure 19.

Again using a smaller learning rate seems beneficial in the first iterations but it worsens the performance as the iterations progress.

## A.7 WARM START VS COLD START

We corroborate the findings of Beck et al. (2021) that especially in the first iterations retraining the model at every cycle improves the overall performance for most AL strategies and that the difference between cold and warm start diminishes in the later AL rounds. Surprisingly LLL4AL benefits from updating the backbone and shows worse performance when compared to the other AL strategies.

Note that however this findings do not generalize over different datasets. In fact in figure 21 we see that a warm-start approach is to prefer when using the SVHN dataset for all the methods under investigation and that no significant difference between the two approaches can be observed on FashionMNIST, beside LL4AL that performs slightly worse compared to the other methods with the cold-start approach. Figure 22 shows the performance of the strategies when selecting for LL4AL a Warm-start approach while for the others Cold-start. It emphasises the importance of choosing the best possible configuration for a specific AL strategy, since here neither LL4AL outperforms the other benchmark strategies nor the contrary is true.

## A.8 BATCH SIZE (NUMBER OF QUERIED SAMPLES IN EACH AL CYCLE)

In this section we want to analyze the effect of the AL batch size (the number of queried samples in each AL cycle). For this matter, we run the same experiment with three different batch sizes: 100, 1000 and 2000. In all of them, we started with 1000 labeled data points and stopped when more than 10000 were labeled. The results are reported in figure 23.

Using a warm-start approach combined with a small batch size improves the overall accuracy of all AL strategies, especially in the first iterations, as can be seen from figure 24.

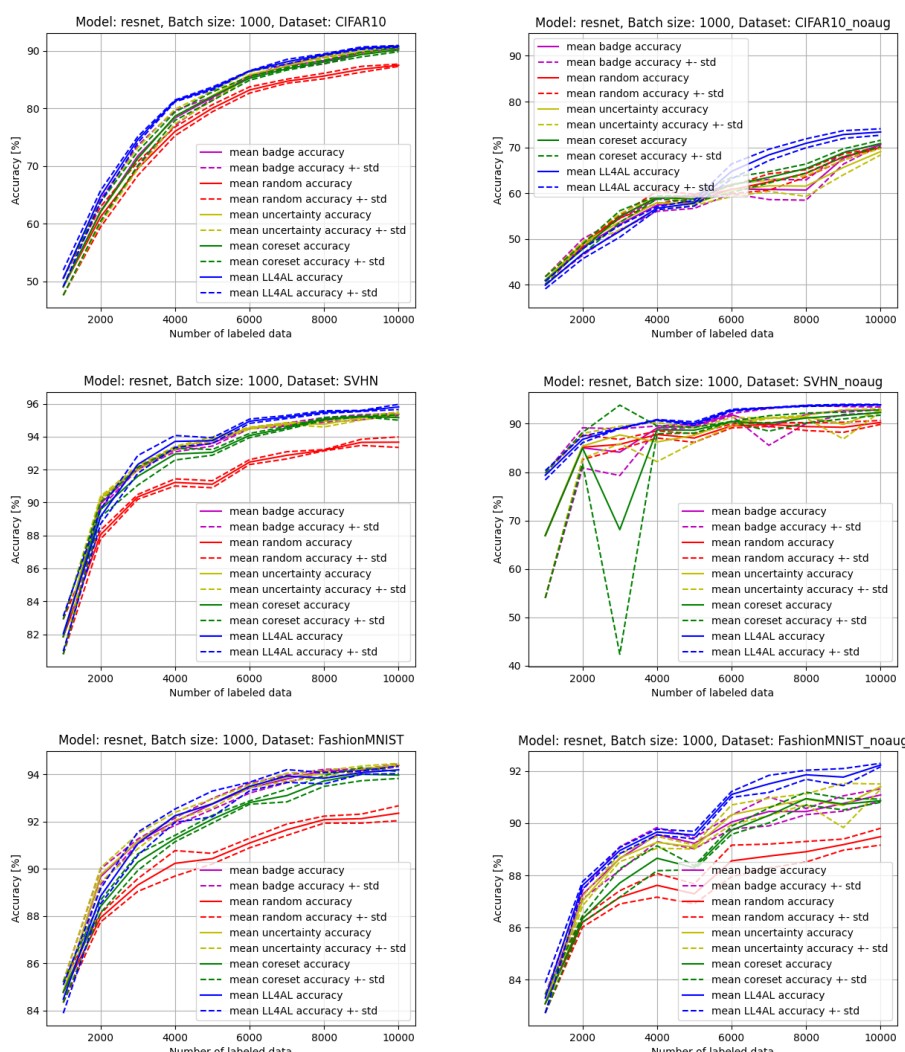

Figure 14: Results when using data augmentation (left) vs no data augmentation (right) on different datasets

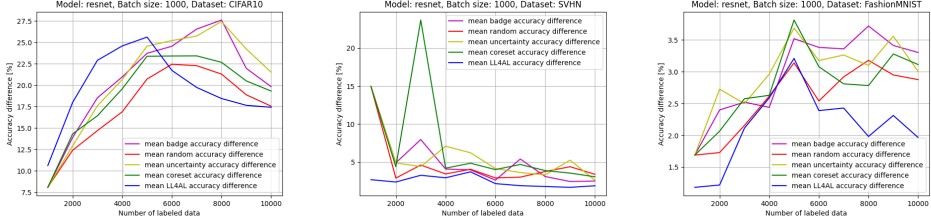

Figure 15: Accuracy difference between using data augmentation vs not using it for different datasets

## A.9 USE OF LABELED AND UNLABELED DATA

Figure 25 shows a comparison between the performance of the AL strategies when all methods train jointly also the discriminator compared to their performance when only the backbone is trained. Similarly figure 26 shows the same for the WAAL method.

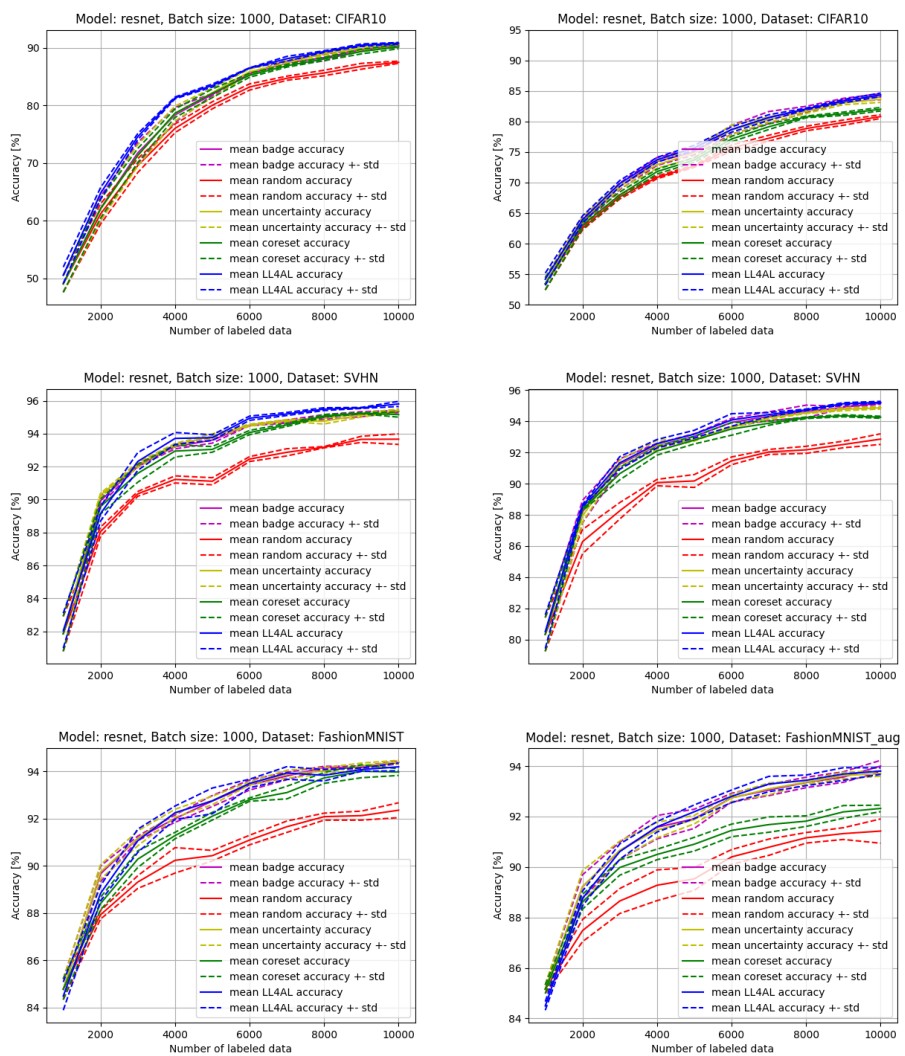

Figure 16: Results when using SGD optimizer (left) vs Adam optimizer (right) on different datasets

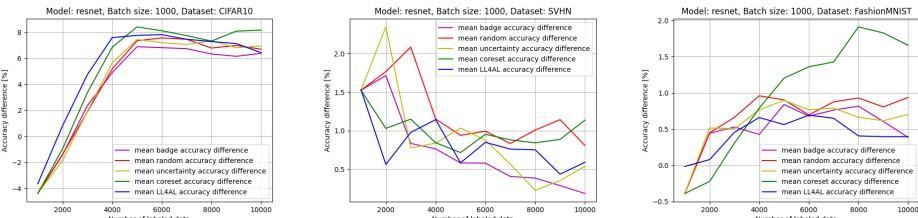

Figure 17: accuracy difference between the default settings using SGD optimizer and the results using Adam optimizer for different datasets

## A.10 NETWORK ARCHITECTURE/ LOSS FUNCTION

Figure 27 shows a comparison between the performance of the AL strategies when all methods train jointly also the loss module compared to their performance when only the backbone is trained.

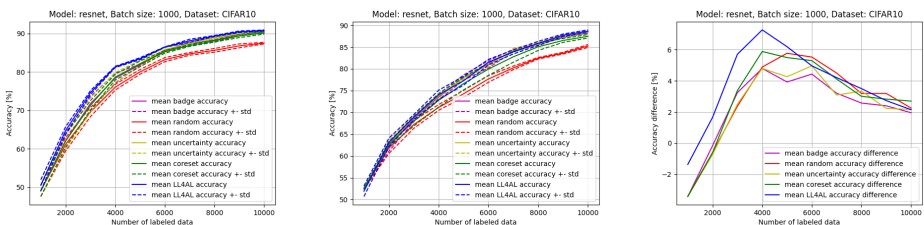

Figure 18: $LR = 0.1$ (left), $LR = 0.01$ (center), accuracy difference between the two (right)

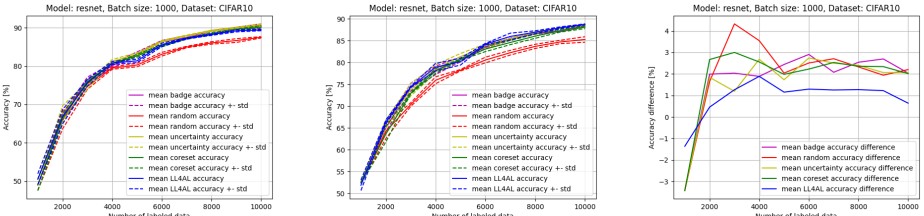

Figure 19: Cold-start: $LR = 0.1$ (left), $LR = 0.01$ (center), accuracy difference between the two (right)

## A.11 INITIALIZATION

### A.11.1 INITIAL TRAINING SET

From figure 28 we see that in the first iteration we have a $2\%$ points accuracy improvement for all the methods beside LL4AL, and that the benefit rapidly vanishes with the following AL iterations. The results depicted in figure 29 show us that on SVHN the samples selected with Kmeans++ with the Imagenet features worsen the performance in the first iteration.

Figure 30 shows an accuracy deterioration of more than $10\%$ points in the first AL cycle using the initial labeled pool selected with K-center.

From figure 31 wee see that K-center initialization performs worse than random also with SVHN. In this case the accuracy drop is smaller compared to the one observed with CIFAR10.

### A.11.2 UNSUPERVISED PRETRAINING OF THE NETWORK

Results when using unsupervised pretraining of the network on CIFAR10 are shown in figure 32 and on CIFAR100 in figure 33.

### A.11.3 SUPERVISED PRETRAINING OF THE NETWORK

Preliminary results, showed that sing a downloaded pre-trained (on ImageNet) ResNet 18 slightly worsen the overall performance of the AL methods. This probably because of the higher resolution of ImageNet. Since pre-training ResNet18 on a low resolution ImageNet is very time consuming we opted to pre-train ResNet-18 on the TinyImageNet dataset and used it as our initial model at every AL cycle. As figure 34 shows, better performance is achieved using the supervised pre-trained network, compared to randomly initialized network.

## A.12 LABELING EFFICIENCY

Usually, when new AL methods are developed, the results are shown by comparing the test accuracy with respect to the number of labeled points for the new methods and for existing benchmarks. This allows to highlight the benefit of a newly proposed strategy compared to the baselines, but it does not give explicitly the reduction of annotated samples that a new sampling strategy needs to achieve

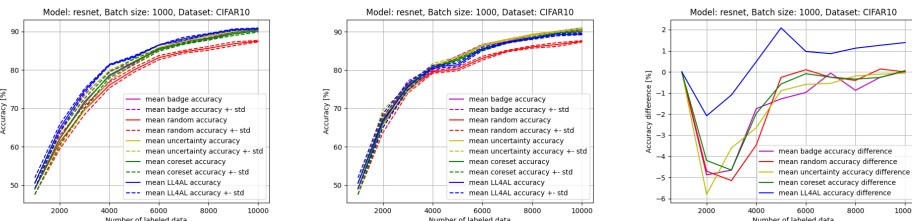

Figure 20: From left to right: warm start, cold start and their accuracy difference

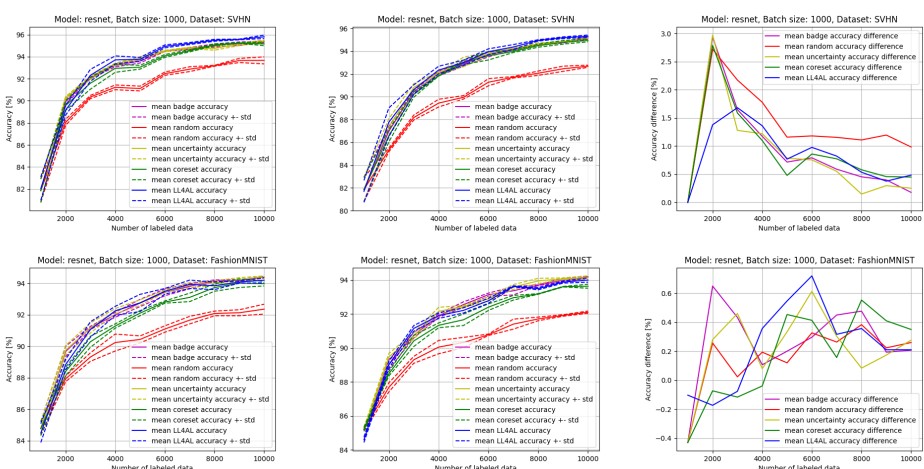

Figure 21: From left to right: warm start, cold start and their accuracy difference on SVHN (top) and FashionMNIST (bottom)

the same test accuracy as random sampling, which is the main goal of every AL strategy. Hence, we show in figure 35 the labelling efficiency of different AL methods using the default setting. The labeling efficiency for a given test accuracy is defined as the ratio of the minimum number of labeled images needed by random sampling divided by the minimum number of images needed by the investigated AL strategy, to reach a given accuracy. We conclude that all AL methods are label efficient across most datasets.

## A.13 CHOICE OF THE LEARNING RATE FOR THE OPTIMIZER COMPARISON

To support our choice of comparing the optimizers using LR of 0.1, we present in figure 36 the results of the different AL strategies when using Adam or SGD as optimizers with a learning rate of 0.1 or 0.01. Comparing the results, we can say that overall the methods reach a lower accuracy when using Adam optimizer combined with a LR of 0.01 compared to the methods when they use SGD and a LR of 0.01. In addition using Adam combined with a LR of 0.01 reaches slightly lower accuracy for all the methods compared to Adam and LR 0.1, making the latter the more appropriate choice when comparing different optimizers. Concerning ranking shifts, we can see that LL4AL is only the top performer when using SGD combined with a LR of 0.1, while the ranking between the other methods stays the same.

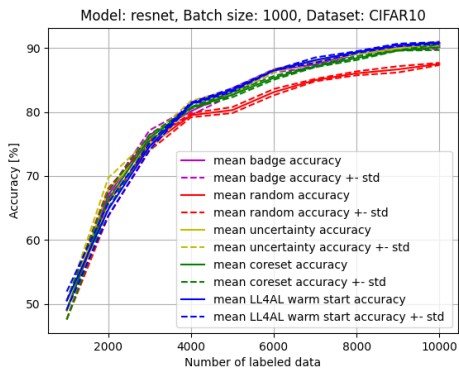

Figure 22: LL4AL Warm-start others Cold-start

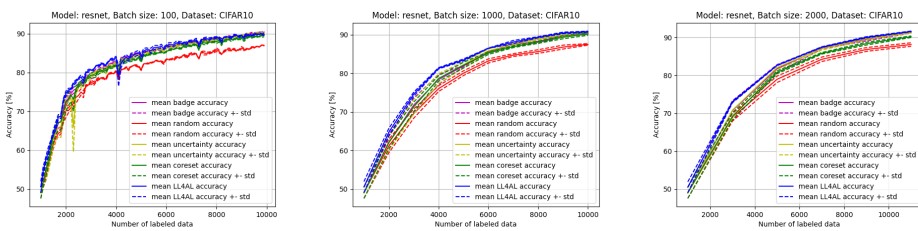

Figure 23: Results for different batch sizes on CIFAR10. From left to right: 100, 1000, 2000 samples queried respectively.

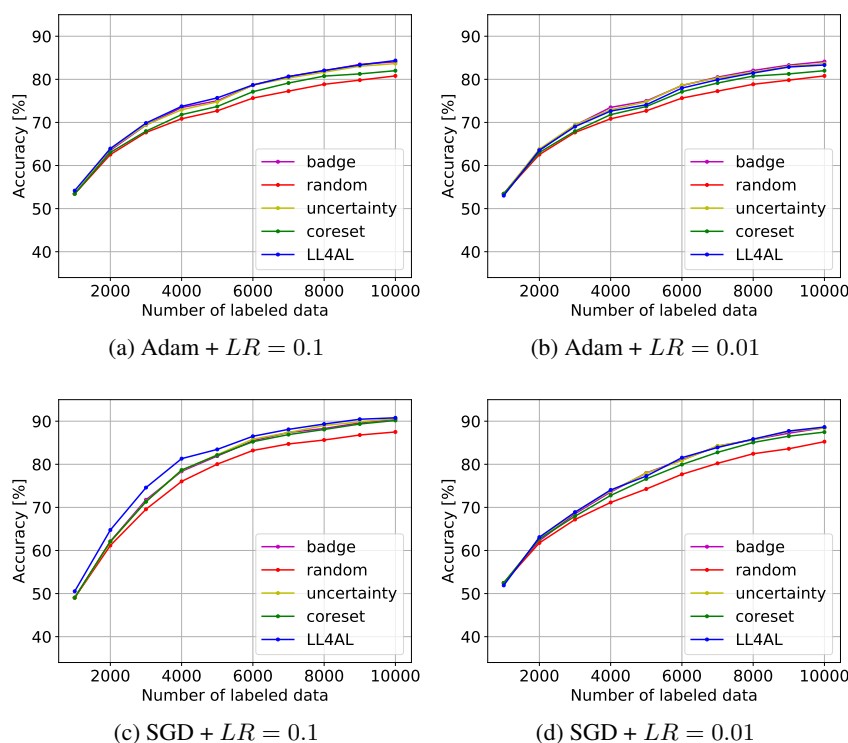

Figure 36: Comparison between different optimizers and learning rates for several AL methods on CIFAR10.

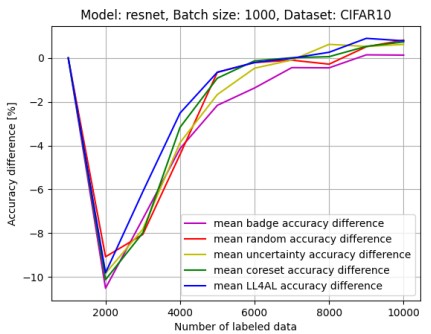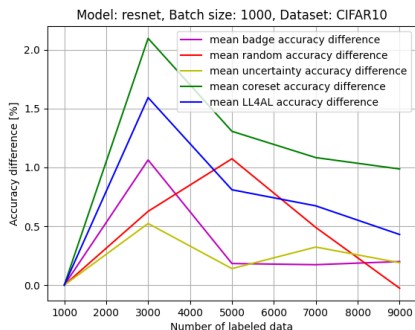

Figure 24: Accuracy difference between a batch size of 1000 with one of 100 (left) and a batch size of 1000 with one of 2000 (right)

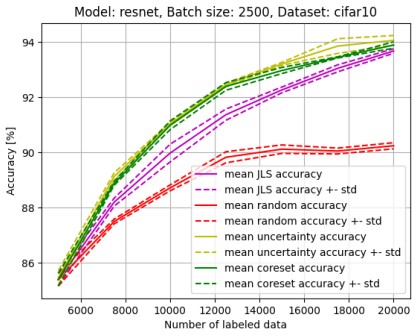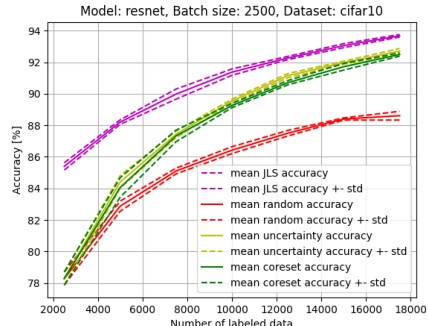

Figure 25: Results when jointly training the discriminator network for all methods vs only for JLS

## A.14 STARTING POINT

There is no consensus across the literature on how many labeled data-points should be used for the first AL cycle when reporting the results. Figure 37 shows how methods that rely on a properly trained backbone, such as Coreset which uses its resulting feature representations of the images for the acquisition, or the Uncertainty method which relies on its output probabilities, perform similarly to Random sampling when the amount of labeled data is extremely small, while methods such as LL4AL which learns jointly the additional task of the loss module performs significantly better than Random sampling also for the backbone trained on 200 images. On the contrary when increasing by a factor of 5 the starting amount of labeled data no significant difference in the ranking of the methods is seen compared to the default configuration.

## A.15 REDUNDANCY

Redundancy is achieved by modifying the training set, such that there is a chosen amount of unique data-points and the remaining ones rest are repeated a given number of time. In our experiment we chose to have 6000 unique data-points and duplicate the remaining images 2, respectively 20, times. Figure 38 shows the respective results. We corroborate the results of Beck et al. (2021) that the diversity based approaches (BADGE and Coreset) are more robust to redundant dataset compared uncertainty based approaches (Uncertainty and LL4AL). Since in our framework the Uncertainty method is performed o a random subset of the unlabeled set we still outperform Random sampling also in the case of a 20× redundant dataset. For LL4AL and Coreset no experiment were presented and therefore a comparison between our and their results is not possible.

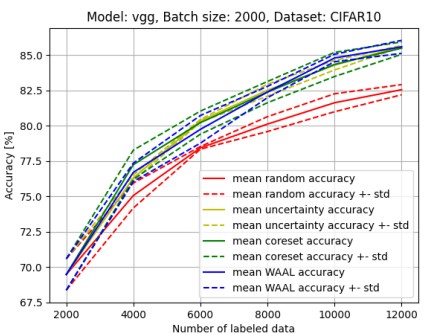 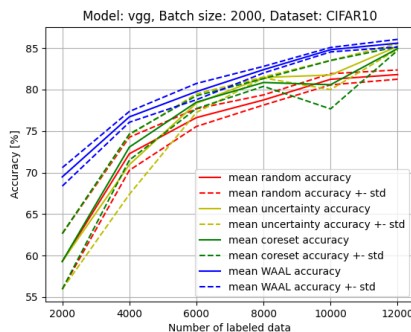

Figure 26: Results when jointly training the discriminator network for all methods vs only for WAAL

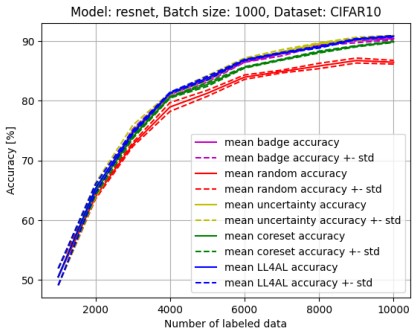 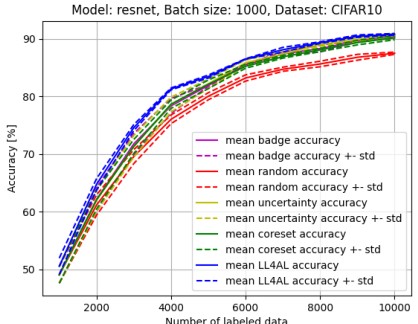

Figure 27: Results when training for all methods the loss module vs only for LL4AL

### A.16 SUBSET SELECTION

Uncertainty and LL4AL are acquisition strategies that are applied on a random subset of the unlabeled data as it is proposed in (Beluch et al., 2018) and (Yoo & Kweon, 2019). This subset has been chosen to be $10\times$ the number of queried data. In this subsection we investigate the influence of selecting a subset not randomly, but using diversity based query strategies, such as BADGE and Coreset, to see if the uncertainty based methods benefit from a diverse subset. The results are shown in figure 39.

The proposed methods perform almost en-par with their benchmarks. The most relevant results are the accuracy difference between the modified Uncertainty, respectively LL4AL, and the default version (green and blue lines in the accuracy difference plots).
The benefit of this subset selection can be seen best when the diversity of the subset is critical, for example in the case of a redundant dataset. Since LL4AL and Uncertainty performed worse than Coreset and BADGE on the $20\times$ redundant dataset, we rerun the same experiment with the improved subset selections. The results are shown in figure 40, where we see how LL4AL for both subset selection methods outperform all benchmarks, in particular BADGE and Coreset. There is a significant improvement also for Uncertainty, which now performs en-par with BADGE and Coreset.

### A.16.1 RANDOM SUBSET IS BENEFICIAL?

A natural question that arises is whether the random subset selection is really beneficial for the Uncertainty method. Figure 42 shows a comparison of Uncertainty applied on the complete unlabeled dataset and only on a random subset of it. Figure 43 shows what happens when we change the starting point and the AL batch size for Uncertainty on the full unlabeled set. For the case where we

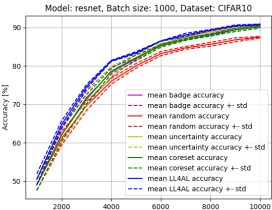 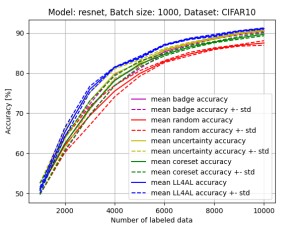 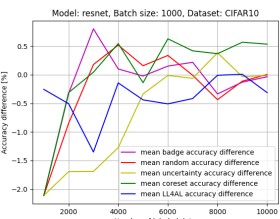

Figure 28: From left to right: reference result, result with Kmeans++ initialization, accuracy difference between the two strategies on CIFAR10.

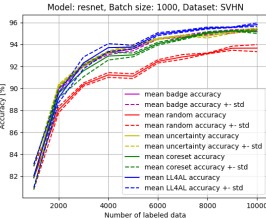 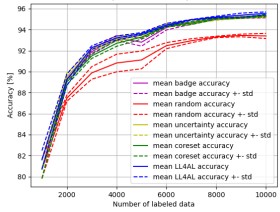 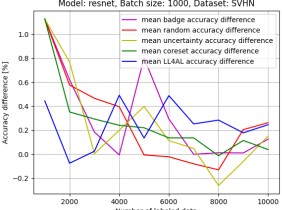

Figure 29: From left to right: reference result, result with Kmeans++ initialization, accuracy difference between the two strategies on SVHN.

start with 5000 labeled data and add 2500 at each iteration, we see that Uncertainty without subset selection outperforms by a large margin all benchmark methods and sets a strong benchmark that new proposed AL strategies should try to beat when considering this range of labeled data.

## A.17 VAAL

The results for VAAL have not been consistent across the AL literature. Some papers, such as (Sinha et al., 2019; Zhang et al., 2020; Kim et al., 2021), find that VAAL outperforms Random sampling by significant margin, while others, such as (Gudovskiy et al., 2020; Caramalau et al., 2021a) find that VAAL achieves comparable performance as Random sampling. We have reproduced the results of VAAL using their codebase finding a comparable accuracy curve as presented in the paper. However, the results for Random sampling using their codebase were significantly better than the reported ones and comparable to the results of VAAL. We also adapted their code to our framework, showing a significant improvement in the overall accuracy ($\geq 15\%$). However, comparing the best result for VAAL with the best result for Random sampling in our codebase we found again that the two methods perform en-par. In detail our implementation differs in the following: We separated the training of the backbone from the training of the sampler; we trained the backbone with the same settings as for the other method; we got rid of the validation set, accounting for $10\%$ of the whole dataset, to select the best model, since it is not common to have one in the AL literature (see table) and the trade-off between having additional data to train on compared to use them just for validation goes clearly in the direction of our adopted version.

## A.18 JLS

What is the influence of the main difference hyperparameter settings that our default configuration has compared to the ones of the origial JLS implementation? Figure 45 shows the results thereof.
To clarify the legend we give a small explanation for every curve: JLS standard is the original paper implementaiton; Random standard is Random sampling using their default configuration; Random + train sampler is Random sampling when the backbone is jointly trained with the discriminator; LR_1 refers to a $LR = 0.1$ and LR_2 to a $LR = 0.01$; the order of the appearance of the learning rates is corresponding to the $LR$ of the backbone (if it appears first) or to the $LR$ of the discriminator (if it appears second); JLS NoNesterov means we do not use Nesterov momentum (Sutskever et al., 2013) in the SGD optimizer; JLS Model_update refers to the original implementation but with a

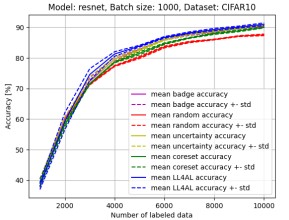 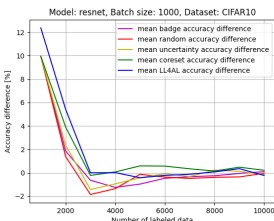

Figure 30: From left to right: reference result, result with K-center initialization, accuracy difference between the two strategies on CIFAR10.

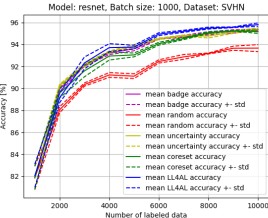 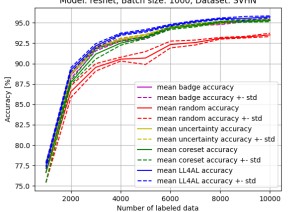 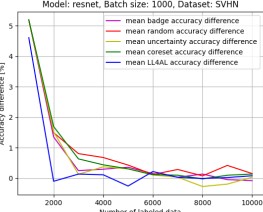

Figure 31: From left to right: reference result, result with K-center initialization, accuracy difference between the two strategies on SVHN.

**Warm-start approach.** Since the best performing variant is the JLS Model_update we adopted that configuration in our codebase.

The results of JLS in our codebase for different datasets can be seen in figure 46. JLS on SVHN (not reported in (Caramalau et al., 2021b)) after the first AL cycle performs worse than the other benchmark methods. The authors reported better accuracy also for FashionMNIST, but chose for this dataset to use an extreme AL case with an initial labeled set consisting of only 100 images and a final one of 700 images.

## A.19 EXTREME AL

Which are the observations we can make when only a handful of samples are used? Figure 47 shows the results for this extreme AL scenario on different datasets. We can see that on more challenging datasets (CIFAR10 and CIFAR100) where the accuracy is relatively poor, there is no consistent gain of the AL strategies over Random sampling. On CIFAR100 LL4AL, Uncertainty and Coreset even underperform Random sampling. On easier datasets, such as FashionMNIST the methods perform slightly better than Random sampling. In general from the dashed lines we see a big variance in the results especially in the first iterations. If the CNN is better trained (unsupervised pretraining), are we going to see better performance for the benchmark methods compared to Random sampling? Figure 48 shows the corresponding experiment.

Even though the performance is much better with unsupervised pretraining (more than $100\%$ accuracy improve in the first cycles and still more than $40\%$ in the last iteration) the benchmark methods perform comparably to Random sampling. Note however that a real improvement over Random sampling is seen only after $75\%$ accuracy in the case where we start with 1000 samples and add 1000 at each iteration. This is also seen in this experiment: when the accuracy is $\geq 75\%$ the methods tend to outperform Random sampling. Still the variance stays high leading to unexpected behaviours like the huge accuracy drops for all methods after subsequent AL cycles.

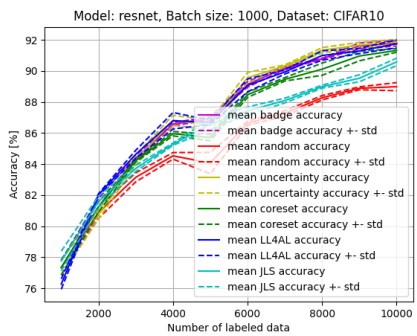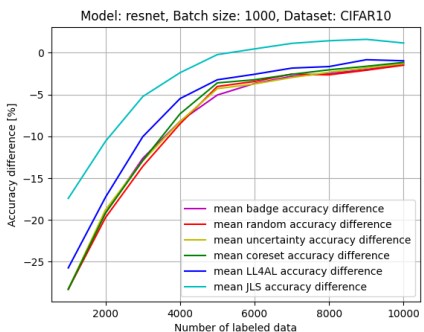

Figure 32: Results for different algorithms when starting with an unsupervised pretrained network at each AL round (left) and the accuracy difference with the default configuration (right)

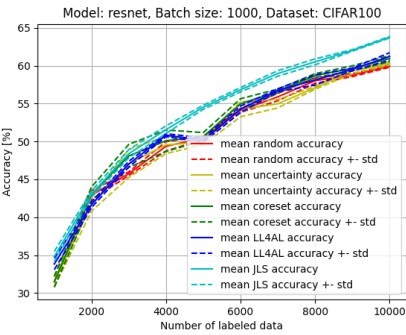

Figure 33: Results for different algorithms on CIFAR100 with an unsupervised pretrained network

## B TABLES

In this section we provide two tables: Table 3 summarize the recent proposed AL strategies, defines the investigated AL setup (AL budget, number of queried samples, etc) and gives the reported accuracy range of the proposed method and of Random sampling. Table 4 provides the implementation details of the methods. Unless otherwise stated, if for different datasets the setup differed too, then the corresponding setups where ordered in the same way as the datasets and separated by a slash. A '-' means that the information was not mentioned. The following abbreviation are used for the AL methods: Coreset for (Sener & Savarese, 2018), VAAL for (Sinha et al., 2019), BADGE for (Ash et al., 2019), LL4AL for (Yoo & Kweon, 2019), JLS for (Caramalau et al., 2021b), WAAL for (Shui et al., 2020), SRAAL for (Zhang et al., 2020), TA-VAAL for (Kim et al., 2021), ISAL for (Liu et al., 2021), FKSSAL for (Gudovskiy et al., 2020), CoreGCN and UncertainCGN for (Caramalau et al., 2021a), VaB-AL for (Choi et al., 2021).

| Method | Diversity vs Uncertainty | Dataset | Acquisition strategy | Starting point | Cycles | Batch size | Accuracy range [%](start-mid-end) | Accuracy range (start-mid-end) random | Fully supervised acc | Code available |
|---|---|---|---|---|---|---|---|---|---|---|
| Coreset | Diversity | CIFAR10/CIFAR 100/SVHN | K-center greedy on feature space of last layer | 10% | 10 | 10% | 60-81-90/27-57-65/68-93-96 | 60-77-90/27-52-65/68-89-96 | - | Acquistion strategy only |
| VAAL | Diversity | CIFAR10/CIFAR100/Caltech-256 /ImageNet | Data for which discriminator is most confident are coming from unlabeled set | 10% | 7 | 5% | 61-77-80/28.5-41-47.5/65-85-88/39-48.5-53.5 | 61-69-74/28.5-39-44/65-81-85.5/39-46-51 | - | yes |
| BADGE | Diversity - Uncertainty | CIFAR10/SVHN /MNIST | K-Means++ on "hallucinated" gradients of final layer (computed using most likely label) | 100 | 500/50/5 | 100/1000/1000 | 10-50-58 (ResNet-18, CIFAR10, 1000)/10-78-84 (VGG-11,CIFAR10,1000) | All methods almost en-par | - | yes |
| LL4AL | Uncertainty | CIFAR10 | Data with highest predicted loss | 1000 | 10 | 1000 | 52-84-90.5 | 51-80-87 | 0.9302 | yes (not official) |
| JLS | Diversity | CIFAR10/CIFAR100/FashionMNIST | Data for which discriminator is most confident are coming from unlabeled set | 10%, 100 (for FashionMNIST) | 7 | 5%/100 (for FashionMNIST) | 81-87-90.5/43-59-66/58-81-84 | 62-70-76/28 -37-44/58-77-81 | - | yes |
| WAAL | Diversity-Uncertainty | CIFAR10/SVHN /FashionMNIST | Combination of an uncertainty term coming from the task model and a diversity term coming from adversarial network, which discriminates between labeled and unlabeled data | 2000/1000/1000 | 7 | 2000/1000/1000 | 55-68-75/75-86-90/73-83-86 | 47-61-72/62-82-87/57-77-81 | - | yes |
| SRAAL | Diversity-Uncertainty | CIFAR10/CIFAR100/Caltech-101 | Discriminator predicts unlabeled data's state value (not only 0/1 but OUI assigns [0,1] value), and the top-K are selected | 10% | 7 | 5% | 80-91-92.5/41-59-67/65-86-90 | 80-88.5-91/41-52-62.5/65-80-86 | - | no |
| TA-VAAL | Diversity-Uncertainty | CIFAR10/CIFAR100 | Discriminator predicts unlabeled data's state value (based on the ranking of the losses) | 1000/2000 | 10 | 1000/2000 | 53-85-90/21-58-68 | 50-82-87/20-55-63 | - | yes, but not complete |
| ISAL | Neither nor | CIFAR10/CIFAR 100/SVHN | Select points with the most positive influence on model performance, which is computed starting by their expected gradients | 1000/5000/2% | 10 | 1000/5000/2% | 45-82-90/37-55-63/38-63-73 | 45-80-87/37-53-31/38-61.5-70.5 | - | no |
| FKSSAL | Diversity | MNIST/SVHN/ ImageNet | Datapoints that maximize Practical Fisher Kernel (minimize distribution shift between unlabeled data and weakly-labeled validation set) | 0.25%/0.25%/5% | 10 | 0.25%/0.25%/5% | 86-97.5-97.8/83-91.5-93.5/42-64-68.5 | 84-94-95.5/83-88-90/42-62.5-64.5 | 98.5%/96%/70% | yes |
| CoreGCN UncertainGCN | Diversity- Uncertainty | CIFAR10/CIFAR100/FashionMNIST/SVHN | Coreset on first layer of GCN and Uncertainty on output of GCN | 1000/2000/1000/1000 | 10 | 1000/2000/1000/1000 | 51-85-90.5/20-57-68/73-93-94/83-90.5-92 | 51-82-86/20-54-63/73-88-92/83-88-90 | - | yes |
| VaB-AL | Neither nor | CIFAR10/CIFAR100 | Datapoints with highest probability of leading to a classification error | 2000 | 6/5 | 2000 | 72-87-91.5/22-48-56 | 72-85-87/22-48-56 | - | no |

Table 3: Description and setup of different AL methods

| Method | Dataset | Backbone | Architecture | Data augmentation | Optimizer | Length of training [epochs] | Learning rate | Initial labeled set | Use of validation set | Cold-Start vs Warm start | Unsupervised pre-training | Use of unlabeled data |
|---|---|---|---|---|---|---|---|---|---|---|---|---|
| Coreset | CIFAR10/ CIFAR 100/ SVHN | VGG-16 | Backbone only | - | RMSProp | - | 0.001 | Random | - | Cold-Start | no | no |
| VAAL | CIFAR10/CIFAR 100/Caltech-256/ImageNet | VGG-16 | Backbone (trained separately) + VAE-Discriminator (minimax training) | Random-Horizontal-Flip | SGD for Backbone, Adam for other nets | 100 | 0.01 | Random | yes | Cold-Start | no | yes, but not for backbone |
| BADGE | CIFAR10/ SVHN/MNIST | ResNet-18/VGG-11/MLP | Backbone only | no | Adam | until training acc > 99% | 0.01 | Random | no | Cold-Start | no | no |
| LL4AL | CIFAR10 | ResNet-18 | Backbone + loss module jointly trained | Random-Horizontal-Flip, Random-Crop | SGD | 200 | 0.1 | Random | no | Warm-Start | no | no |
| JLS | CIFAR10/CIFAR100/FashionMNIST | VGG-16/VGG-16/ResNet-18 | Backbone + discriminator jointly trained | Random-Horizontal-Flip, Random-Crop | SGD | 200 | 0.01 | Random | no | Cold-Start | no | yes |
| WAAL | CIFAR10/SVHN/FashionMNIST | VGG-16/VGG-16/LeNet-5 as feature extractors + two layer MLP for classifier | Backbone + discriminator jointly trained with minimax game | No (but available code uses Random-Horizontal-Flip, Random-Crop) | SGD | 80 + early stopping | 0.01 | Random | no | Cold-Start | no | yes |
| SRAAL | CIFAR 10/ CIFAR 100/ Caltech-101 | ResNet-18 | Backbone (trained separately) + Unified Representation Generator-Discriminator (minimax training) | - | - | - | - | Random and K-center (on features of VAE) | - | Warm-Start | no | yes, but not for backbone |
| TA-VAAL | CIFAR 10/CIFAR 100 | ResNet-18 | Backbone + loss module jointly trained and VAE-Discriminator (minimax trained) | Random-Horizontal-Flip, Random-Crop | SGD for Backbone, Adam for other nets | 200 | 0.1 | Random | no | Cold-Start | no | yes, but not for backbone |
| ISAL | CIFAR10/CIFAR 100/SVHN | ResNet-18 | Backbone only | Random-Horizontal-Flip, Random-Crop | SGD | 200 | 0.1 | Random | - | Cold-Start | no | no |
| FKSSAL | MNIST/SVHN/ImageNet | LeNEt /ResNet-10 / ResNet-18 | Backbone only | no | SGD | 50/ 35/ 60 | 0.05/0.1/0.1 | Random | yes (weakly-labeleld) | Cold-Start | yes | yes, pre-training |
| CoreGCN UncertainGCN | CIFAR10/CIFAR100/FashionMNIST/SVHN | ResNet-18 | Backbone + Graph Convolutional Network trained to distinguish between labeled and unabeld data | Random-Horizontal-Flip, Random-Crop (only for CIFAR10 and CIFAR100) | SGD | 200 | 0.1 | Random | no | Cold-Start | no | yes, but not for backbone |
| VaB-AL | CIFAR10/CIFAR100 | ResNet-18 | Backbone + VAE | - | SGD | 200 | 0.1 | Random | no | - | no | yes, but not for backbone |

Table 4: Implementation details for different AL methods

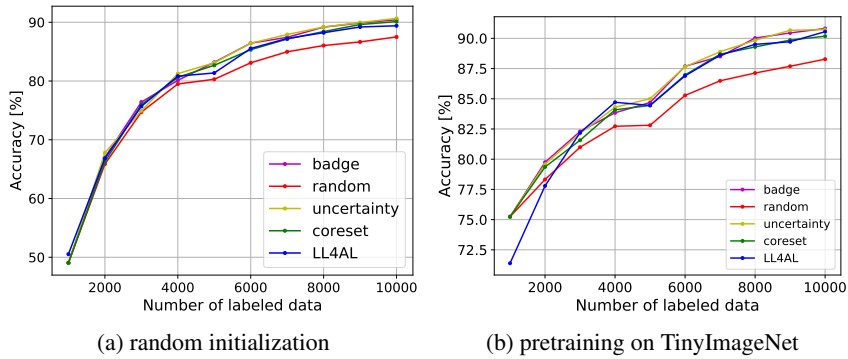

(a) random initialization      (b) pretraining on TinyImageNet

Figure 34: Comparison between randomly initialized network weights (left) and pre-trained weights on TinyImageNet (right).

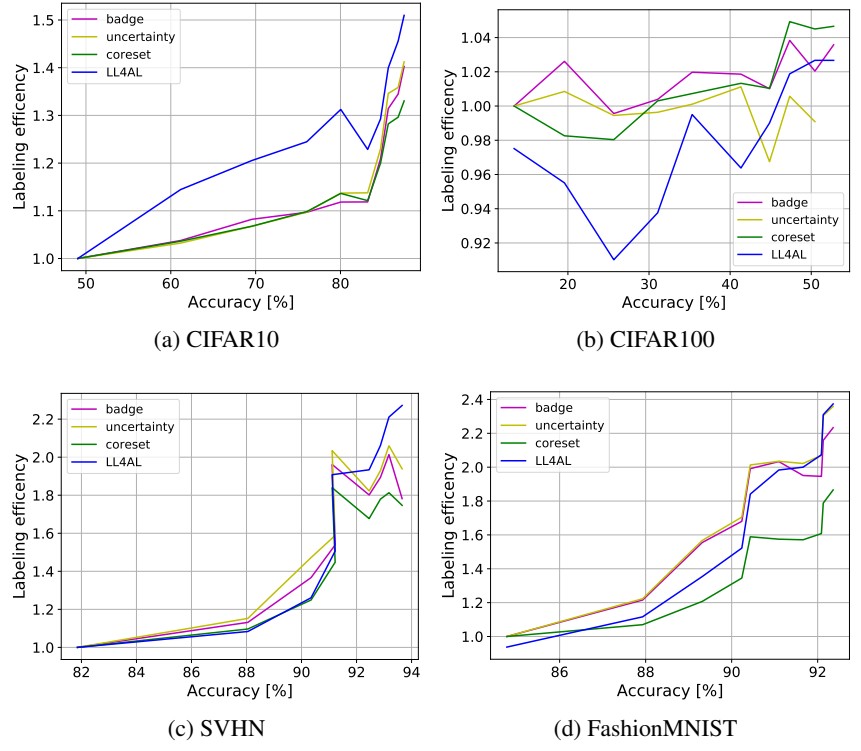

(a) CIFAR10      (b) CIFAR100

(c) SVHN      (d) FashionMNIST

Figure 35: Labelling efficiency on different datasets

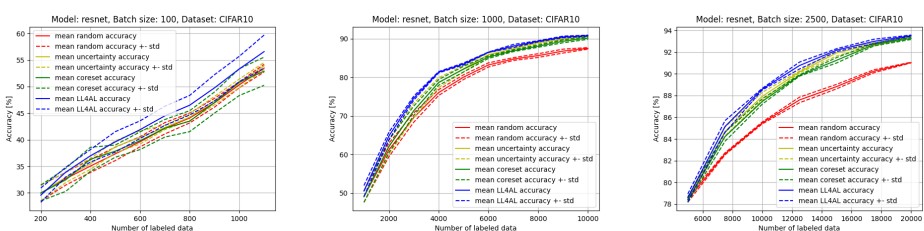

Figure 37: Results for different starting points and number of queried points

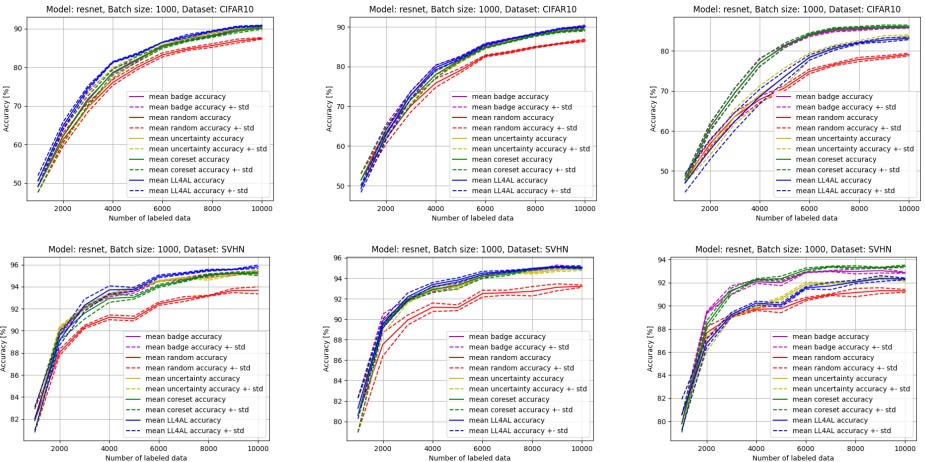

Figure 38: Results with standard datasets (left), $2\times$ redundant datasets (center), $20\times$ redundant datasets (right)

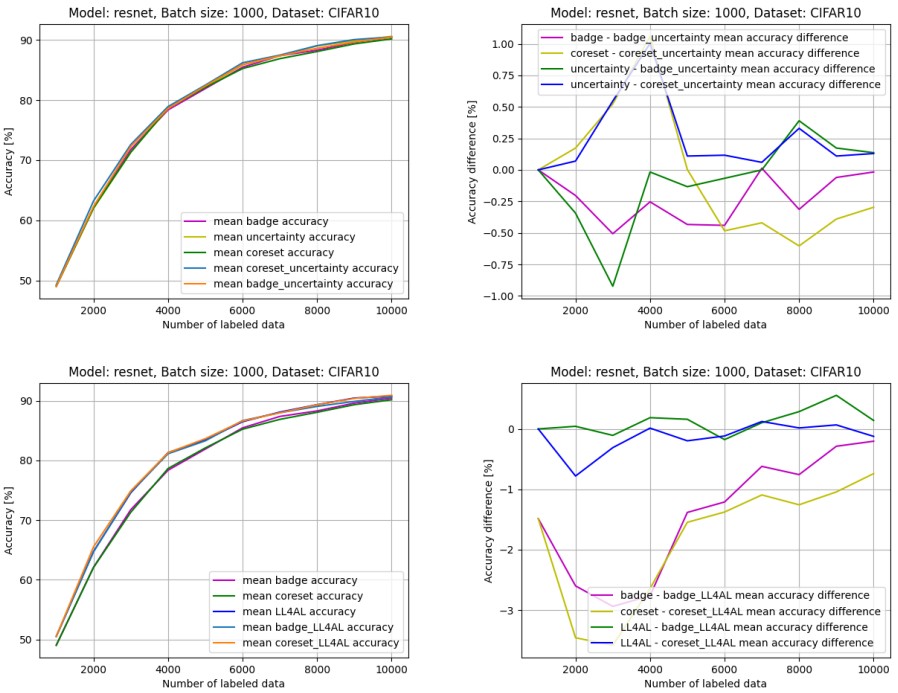

Figure 39: Overall performance of the investigated methods and their benchmarks (left) accuracy difference between the benchmarks and the proposed methods (right)

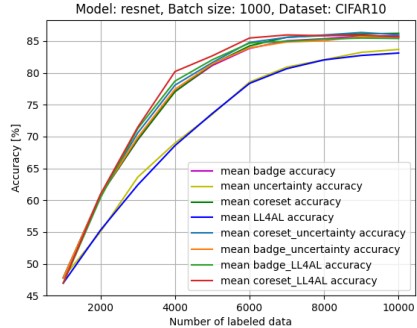

Figure 40: Performance of proposed subset selection methods vs their benchmarks on a $20\times$ redundant CIFAR10 dataset

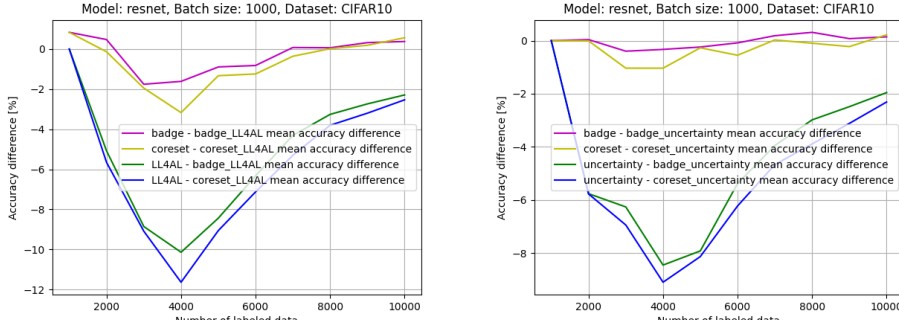

Figure 41: Accuracy difference between proposed subset selection methods and their benchmarks on a $20\times$ redundant CIFAR10 dataset

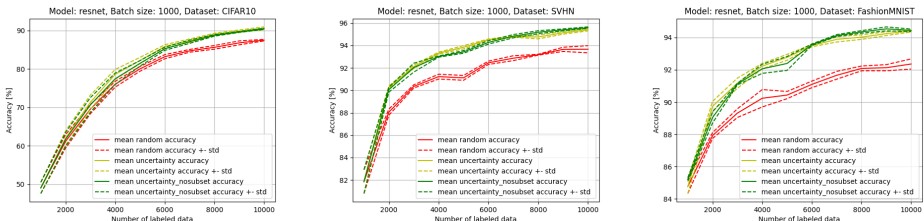

Figure 42: Results for the Uncertainty method with and without random subset selection for different datasets.

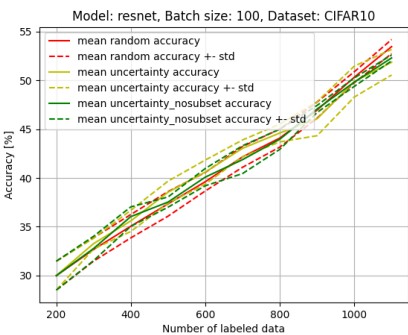
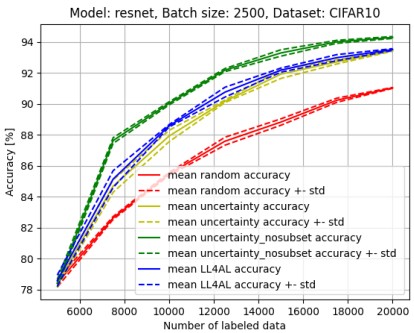

Figure 43: Results for Uncertainty with and without random subset selection for different starting points and AL batch sizes

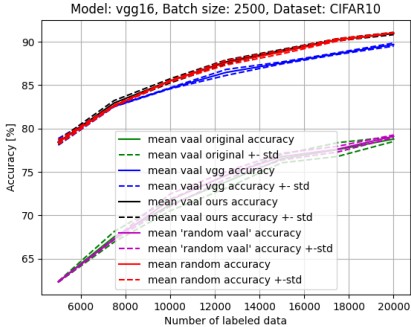
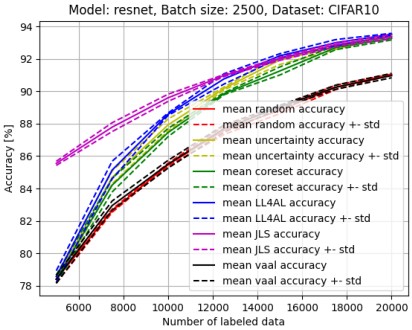

Figure 44: VAAL results: on the left the original implementation vs our implementation, on the right VAAL compared to other methods in our framework

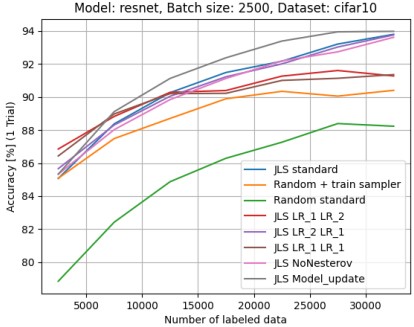

Figure 45: Hyperparameter study on JLS

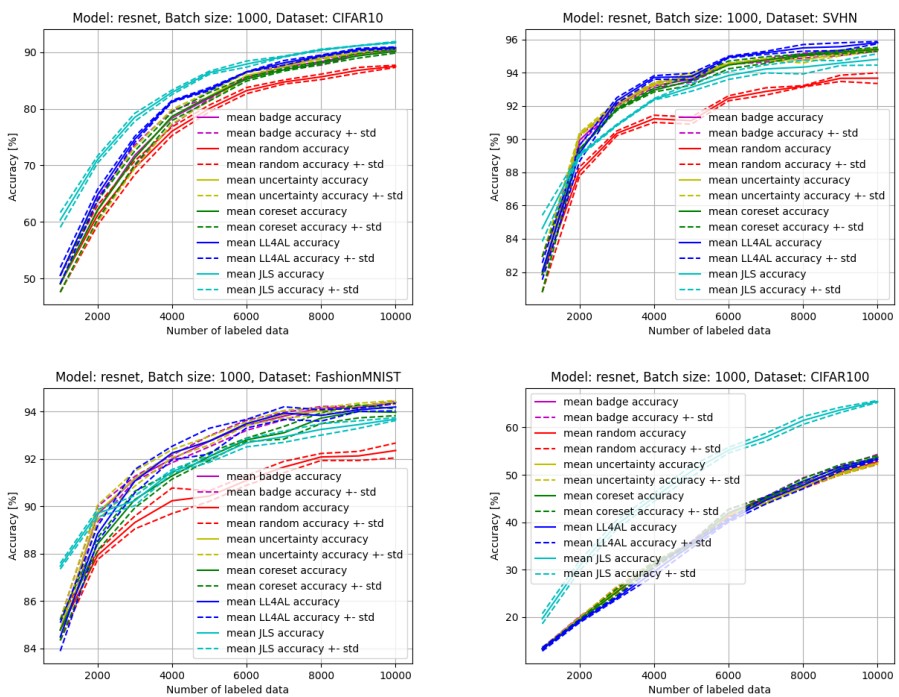

Figure 46: Results of the JLS method on different datasets

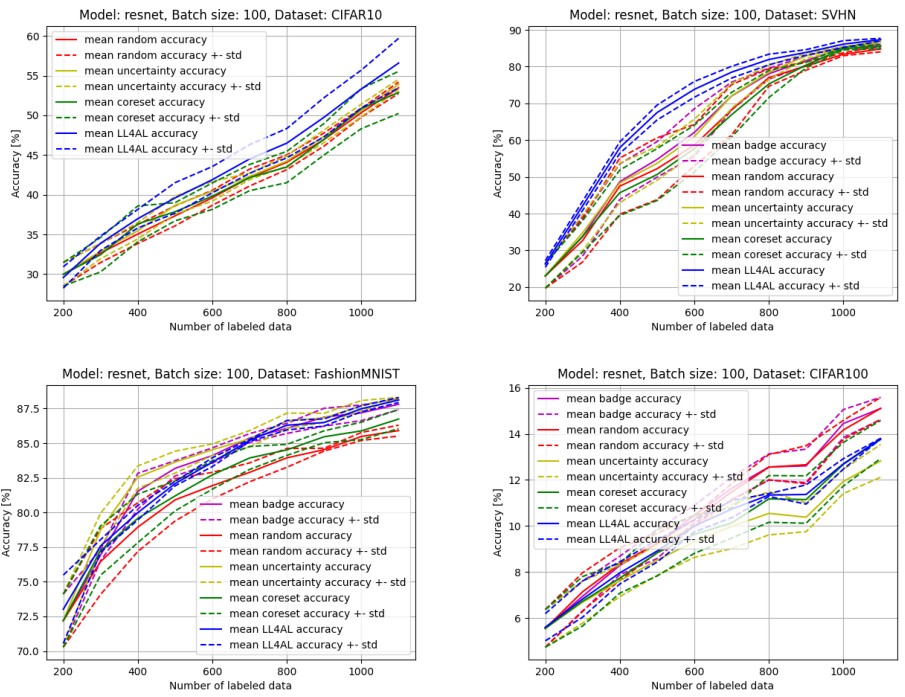

Figure 47: Results of extreme AL for different datasets

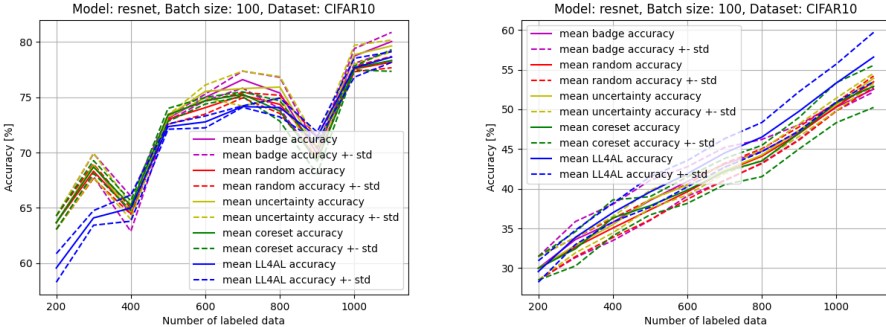

Figure 48: Results for extreme AL with and without unsupervised pretraining

