# OpenReview forum: "Best Practices in Pool-based Active Learning for Image Classification"
_ICLR.cc/2022/Conference — ICLR 2022 Submitted_

### Official Review · Reviewer_ow4N · 2021-11-01

**Correctness:** 3
**Technical Novelty And Significance:** 3
**Empirical Novelty And Significance:** 3
**Recommendation:** 6
**Confidence:** 5

**Main Review:**

Strengths:

- Recent active learning methods show results using different settings, and this paper tackles the important job of evaluating these methods in a consistent manner.

- The analysis w.r.t using additional modules for complementary tasks, and unsupervised pre-training is interesting and useful.

- The paper is clear to understand and the most results are presented effectively.


Weaknesses:

- When comparing different optimizers, the optimal learning rate for different optimizers can be different. Hence, the argument of using the learning rate which is better for one optimizer (SGD), also for the other optimizers (Adam and RMSProp), is not precise. This is especially concerning because the relative ordering of methods might change, and the conclusion w.r.t optimizer might not hold.
- Similar concern as above for experiments with different backbones, as different backbones might need different learning rates to obtain the best performance.

- Other popular methods to use unlabeled data in terms of semi-supervised learning of the classifier should be compared ([1], [2]).

- It would be valuable to see how the performance of these methods would change with datasets having different number of classes, which is another important factor. The three datasets studied here all have 10 classes.

- It would be good to explicitly mention the active learning settings (i.e. initial pool size, budget size) in the experimental setup.

- The effect of supervised pre-training of the network on other larger datasets (e.g imagenet) in section 7 would be useful and make the analysis more complete.


[1]: Liao et. al. Towards Good Practices for Efficiently Annotating Large-Scale Image Classification Datasets
[2]: Mittal et. al. Parting with Illusions about Deep Active Learning

------------------------------------------------------------------------------------------
Some typos:

* Introduction, paragraph 1: it exists a large unlabeled U → there exists a large unlabeled set U
* Introduction, paragraph 2: include the for training → include them for training
* Introduction, paragraph 2: divers way → diverse way
* Section 7, unsupervised pretraining of the network: In such as setting → In such a setting
* Section 7, unsupervised pretraining of the network: Figure 10a show → Figure 10a shows


**Summary Of The Paper:**


The paper provides benchmarking of some of the popular active learning methods on CIFAR10, SVHN and FashionMNIST datasets. Effects of factors such as choice of backbone, data augmentation, optimizers, learning rate, cold vs warm starting are studied and the conclusions are provided as best practices. Analysis is also performed w.r.t using unlabeled data, choosing the initial labeled pool (random vs K-Means++/K-Center), and unsupervised pre-training of the backbone.


**Summary Of The Review:**

I believe more thorough experimentation is needed for backing up conclusions in some parts like the effect of optimizers and backbone. It would be very valuable to compare popular semi-supervised methods in the section on using unlabeled data.

---

> ### Author Response · Authors · 2021-11-19
> **Response to Reviewer ow4N**
>
> Thank you for your valuable comments on our paper. We have carefully addressed them and improved the paper according to your suggestions. We appreciate any further comment if some points can still be improved.
>
> In detail the answers to your concerns:
> 1. We agree with the reviewer that the optimal learning rate for different optimizers but for the same network may differ. We clarified the explanation why we are comparing the optimizers with a learning rate of 0.1 in the main paper. Namely, we evaluated and compared popular optimizers and learning rates used in the literature and we found that a learning rate of 0.1 works best for both SGD and Adam optimizers (see Section A.13 in the supplementary material for the detailed experiments).
>
> 2. Indeed the learning rate required differs from network architecture to architecture. This is why we carefully tuned the learning rate for VGG-16 and ResNet18. However, the purpose of the experiments involving DenseNet was not to benchmark different architecture but rather that choosing a new architecture might require adapting the settings otherwise it might lead to poorly trained networks as you mention that prevent a fair comparison of different AL methods. Typically all AL methods perform similar to random sampling if a classification network is poorly trained. Hence, before comparing methods the overall accuracy of the classification network should be maximized by identifying a suitable training setting.
> 3. Thank you for pointing out theses methods. We agree that our comparison of methods that include unlabeled samples (WAAL and JLS) is by no means complete and that there exist further methods tackling AL from this perspective. The same is of course true for general AL methods. In the end we chose a few popular AL methods to prove our points about evaluation of AL methods. Our intent is not to give a complete summary and evaluation of all AL methods. Nonetheless, we included the mentioned methods in the semi-supervised learning part in the introduction and data section and referenced them.
> Comparing [2] with our results, we observe that for example on CIFAF10 the combination of semi-supervised learning and AL uses the available labeled data more effectively than only the AL does. Similarly it also achieves better results than our version using unsupervised pre-training. The findings of [2], even enforce our conclusion state that it is important to decouple the performance improvement cause by a given training setting from the improvement obtained by a new acquisition strategy (e.g. LL4AL and JLS achieve superior accuracy due to an improved training procedure and losses not due to a better sample selection).
>
> 4. We report results on additional datasets - CIFAR100 (100 classes) and TinyImageNet (200 classes) - in the supplementary material section A.1.
> For CIFAR100 all AL strategies show an improvement over random sampling, even though not as significant as for the smaller datasets containing less categories but more samples per class. For TinyImageNet no method consistently outperforms random sampling. We observed that the investigated AL methods fails when dealing with poorly trained networks since the sample selection procedure is based on outputs of the network. The overall low accuracy on TinyImageNet and the big variance across multiple random seeds indicates that the network is poorly trained and other than the investigate AL methods are required to outperform random sampling in such cases.
>
> 5. We conducted multiple experiments aiming at transfer learning for AL. Preliminary results with a pretrained ImageNet for the baselines showed inferior results than training the model with random initialization. We conclude that the domain gap between the high resolution ImageNet dataset and our low resolution datasets is too big and prevents improving the overall accuracy. Instead, we conducted another set of experiments using TinyImageNet (contains images with a similar resolution as our datasets) to pre-train the classifier. We use these weights in every AL cycle as starting point. Figure 34 in the supplementary shows superior performance of these pre-trained models compared to the randomly initialized ones. Details are included in Section A.11.3 in the supplementary material.
>
> 6. Thanks for bringing our attention to the typos we fixed them in the revised version.

---

### Official Review · Reviewer_Zkvo · 2021-11-02

**Correctness:** 3
**Technical Novelty And Significance:** 2
**Empirical Novelty And Significance:** 2
**Recommendation:** 8
**Confidence:** 4

**Main Review:**

This article is focused on an extensive analysis of popular AL methods and studies the effect of different training settings typically used in the literature. In addition, it discusses the merit of recent trends in AL such as using unlabeled data, initial set construction, and unsupervised pre-training of the network. The work is topical and on a topic that should be of interest to readers. There are a few parts of the article that might benefit from some attention in revision:
The greatest concern is with the assessment of classification accuracy. The article is lacking detail on this important issue. Most AL works (Wei et al., 2015; Sener & Savarese, 2018; Ash et al., 2019; Killamsetty et al., 2021) present their results comparing the test accuracy with respect to the number of labeled points, to highlight the benefit of the selection algorithm compared to their baselines. In contrast, this paper does not explicitly show the labeling efficiency of selection algorithms with respect to random sampling. In addition, it is not clear how many fewer labels a selection algorithm needs in order to achieve that test accuracy with respect to random sampling. I doubt this problem can be removed by the use of labeling efficiency plots and motivate the need for studying labeling efficiency in future works as a means to more holistically understand AL selection algorithms.
I suspect that the codebase of this article would be a useful resource for others to build on this work and help give this article impact on the community.


**Summary Of The Paper:**

This paper conducts an extensive analysis of state-of-the-art Active Learning (AL) methods (Coreset, BADGE, LL4AL, JLS, and WAAL) and studies the effect of different training settings (Backbone architecture, Initializing backbone weights, Optimizer, and learning rates, with and without data augmentation) and their effect on AL evaluation for image classification. It also highlights the main factors that can influence the performance of AL methods: the construction of an initial training set following a certain strategy instead of random sampling, and pretraining the backbone of the network in an unsupervised manner. In addition, it provides solid benchmarks to compare new with existing methods in sections 5 and 6.

**Summary Of The Review:**

This paper reproduces state-of-the-art results of popular AL methods with different datasets and settings typically used in the literature. It also highlights some approaches (sections 5, 6, and 7) that can influence the performance of AL methods by comparing only test accuracy with respect to a number of labelled points. It is not clear how many fewer labels a selection algorithm needs in order to achieve that test accuracy with respect to random sampling. They also provide a new Pytorch codebase that will allow future researchers to evaluate and compare different strategies in a fair way. This article lacks details on the labeling efficiency of different methods and their quantitative evaluation as done in Beck et al., 2021.

---

> ### Author Response · Authors · 2021-11-19
> **Response to Reviewer Zkvo**
>
> Thank you for your precious comments on our paper. Instead of including label-efficiency plots we decided originally to follow prior work such as (Sinha et al, 2019, Yoo et al., 2019; Zhang et al. 2020, Caramalau et al., 2020) and simply included the accuracy plots showing the achieved test accuracy for a given annotation budget for each method. Nonetheless, we thank the reviewer to draw our attention to the label efficiency topic. We think that while they use the same underlying results as the accuracy plots they can indeed shed the light differently on AL method performance. Hence, we include label-efficiency plots in the supplementary material in Section A.12 of the revised manuscript. The overall conclusions remain the same and are included in the main paper. Please let us know if label-efficiency plots for other than the default setting are needed.  We agree that the codebase will hopefully be of great help for the AL community and will simplify and accelerate the development of new AL methods and ease the comparison with the state of the art.

---

### Official Review · Reviewer_3Uts · 2021-11-02

**Correctness:** 4
**Technical Novelty And Significance:** 1
**Empirical Novelty And Significance:** 3
**Recommendation:** 6
**Confidence:** 3

**Main Review:**

The experiments are extensive and valuable for the better understanding the existing strategies of training Active Learning models. Compared to previous codebase, this work seems to show some different observations and insights. Meanwhile, the conclusions about  various factors in AL methods are insightful and useful for guiding the development of AL strategies.

Some concerns:
1. Is it possible to extend these experiments into more real and complex image datasets, like ImageNet?
2. The conclusions or observations are straightforward, but scattered. Is it possible to provide an overview of the importance of various setting, like training setting, dataset configuration?
3. It would be better to discuss the potential research direction, considering the properties of existing AL methods and datasets.



**Summary Of The Paper:**

This paper provids a codebase for fair comprisons of existing Active Learning methods, and performs a lot of comparative studies on the various influence factors for Active Learning. Some new observations are yielded and help figure out the optimal practices of evaluating AL methods.

**Summary Of The Review:**

Overall, the experiments are extensive and produce many findings that could help understanding of existing pool-based AL methods. The new codebase could provide solid benchmarks for making fair comparisons. Although some adjustments may make a better contribution, this paper is marginally above the acceptance threshold.

---

> ### Author Response · Authors · 2021-11-19
> **Response to Reviewer 3Uts**
>
> Thank you for your appreciation of our work. We addressed your concern carefully. Please let us now if we can clarify other issues.
>
> Responding to your concerns:
>
> 1. We added results on the TinyImageNet datasets in the supplementary material section A.1, where you can find also result for CIFAR100. For CIFAR100 all AL strategies show an improvement over random sampling, even though not as significant as for the smaller datasets containing less categories but more samples per class. For TinyImageNet no method consistently outperforms random sampling. We observed that the investigated AL methods fails when dealing with poorly trained networks since the sample selection procedure is based on outputs of the network. The overall low accuracy on TinyImageNet and the big variance across multiple random seeds indicates that the network is poorly trained and other than the investigate AL method should be deployed to outperform random sampling.
> 2.  We adapted the conclusion following your suggestion.
> In particular we state that using data augmentation and SGD combined with a learning rate of 0.1 provide higher accuracy for all AL methods using ResNet-18 across the datasets. We also bring the attention to the importance of separating the performance of a particular training setting from the acquisition strategy of the AL method.
> 3. We added potential research direction in the conclusion section. We added potential research direction in the conclusion section. In particular we added:
> "Future AL methods should try to clearly disentangle the performance improvement obtained by changes in the classification network and by the proposed acquisition strategy. When evaluating a new method, it should be made sure that the existing state-of-the-art AL strategies used as benchmarks achieve comparable results with the original accuracy reported on a specific dataset and AL interval. Deviating from the standard settings for a novel method complicates a fair comparison. Nonetheless, in such cases a new method should only compare to the baselines that use the novel settings instead of the standard ones only if they achieve better results."

---

### Official Review · Reviewer_k1Y4 · 2021-11-02

**Correctness:** 3
**Technical Novelty And Significance:** 2
**Empirical Novelty And Significance:** 3
**Recommendation:** 5
**Confidence:** 3

**Main Review:**

Pros:
1) The paper can be used as a guidance to utilize AL methods.
2) The conclusions drawn in the submission can be used to design and improve the performance of AL methods.

Cons:
1) The paper is more like a technical report where different experiments are conducted and corresponding results are reported. Minimal analysis or insights are given to discuss why the results are obtained. For example, the authors give a conclusion where LL4AL should use a warm start. But what exactly affects the warm start to be effective for different AL methods and datasets? What results could be expected on other datasets that are not included in this paper?
2) Can the conclusions be generalized to other AL methods? If so, it would be interesting to see the authors using the conclusions to construct an AL system to achieve better performance.

Minor:
1) The font size is too small for the figures to be read. Specifically, please enlarge the font size in Fig. 1,2,3,4,5,7,8.
2) Please consider to put different shapes or use different line types for the methods in comparison for better presentation.

**Summary Of The Paper:**

In this paper, the authors provided a study on how to effectively utilize the existing active learning (AL) methods. Specifically, the authors conducted experiments on the performance of different AL approaches with different training strategies and data manipulations.


**Summary Of The Review:**

To sum up, the paper can be employed as a practical guidance for researchers to conduct AL for image classification. However, minimal analysis or insights are given for the results to be generalized to other datasets or methods. Thus, my initial rating on this submission is borderline reject.

---

> ### Author Response · Authors · 2021-11-19
> **Response to Reviewer k1Y4**
>
> Thank you for your valuable comments on our paper. We have improved the paper according to your suggestions. In particular we reformulated the subsection about warm and cold start, to address the raised concerns. Furthermore, we added a more detailed conclusion where we state the main findings and how they generalize across datasets and methods.
> In addition, we revised the figures. Please note, that the intent of this paper is providing a solid starting ground for future AL methods to simplify comparison with the state of the art and providing insights about the effects that certain settings have on the AL performance. We hope that this inspires future work and that providing a code base supporting the most popular methods and datasets will accelerate AL research. We hope that we could addressed your concerns adequately. Please let us know otherwise.
>
> In detail:
> 1. First, we want to clarify the intent of this paper. Currently a fair comparison between AL methods is very complicated and often lacking in the literature. Different settings than the original are often applied for related methods that then suffer from a drop in performance. Hence, in this paper we first identify such settings that can impact the ranking between different AL methods and verify their impact. Finally, in order to overcome the current shortcomings we propose best-practices for evaluation of AL methods to enable a fair comparison. A good example is warm-start. We observe that LL4AL benefits from warm start across all dataset. Conversely, all other methods achieve higher performance with cold-start on CIFAR10 but lower on SVHN and no significant difference on FashionMNIST. Hence, instead of comparing all methods in the same setting (with warm-start or with cold-start) we propose to simply use the setting for related methods where they work best. If they benefit from the same settings as the proposed method then they should be adopted or otherwise left unchanged. This is important to ensure that the improvement of an AL methods originate from a superior sample selection strategy instead of from improved settings.
> However, why warm-start is beneficial for a specific method or dataset while it is not for others, is beyond this paper since we focus on best practices of AL evaluation.
> 2. As pointed out earlier our intention is not to propose training settings that are suitable for any AL method but rather best practices to evaluate AL methods to enable a fair comparison with the state of the art.
> Nonetheless, we found a training setting that leads to the best performance for the investigated AL methods and since they are shared among all methods the comparison is fair. In particular, using a ResNet18 trained with SGD with a learning rate of 0.1 and using data augmentation leads to state-of-the-art performance on different datasets for all methods. Indeed, the used settings improve the numbers reported in the corresponding papers of Coreset and Badge as well as the the result of common baselines such as Uncertainty sampling. Moreover, we show that unsupervised pre-training leads to an AL system that achieves the best accuracy for a given annotation budget outperforming models trained from scratch considerably.
>
> Minors:
> 1. Thank you for your suggestion, we agree that larger font size for the mentioned figures are indeed necessary. We adapted them accordingly. Moreover, we would like to point out that we include even larger figures in the supplementary material containing additional details such as the standard deviation beside the reported mean in the main paper. Please have a look and let us know if further changes are necessary.
> 2. Following your suggestion, we changed the shape for Fig 8 (a) and (b).

---

### Author Response · Authors · 2021-11-19
**List of changes**

Thanks to all the reviewers for the valuable suggestions and comments. They helped us to improve the quality of the paper.
In particular we changed:

- the last paragraph of the introduction (before the contributions)
- we added the active learning settings in the experimental setup
- we reformulated the conclusions of the "Initializing Backbone Weights" section
- we specified the reasons of our choice of the learning rate in the optimizer comparison in the "Optimizer and Learning Rates" section
- we updated the figures font size
- we reformulated the conclusions
- we added results on TinyImageNet in the supplementary material (section A.1)
- we added a supervised pre-training section in the supplementary material (section A.11.3)
- we added a section about labelling efficiency in the supplementary material (section A.12)
- we added a section about the choice of the learning rate for the optimizer comparison in the supplementary material (section A.13)

---

### Decision · Program_Chairs · 2022-01-20

**Decision:**

Reject

**Comment:**

The authors study the training settings that may affect active deep learning performance, including code/warm start, leveraging unlabeled data, and initial set selection, for each active learning strategy. The findings on several data sets help understand AL more, with some pieces of insights to inspire future research.

The reviewers were at best lukewarm about the work prior to the rebuttal. Some turned more positive but none were willing to strongly champion for the paper's acceptance, even after the authors provided a decent rebuttal. This leaves the paper to be a borderline case, and the recommendation comes from carefully checking the latest revision and calibrating its score with other submissions.

The reviewers are generally positive about the breadth of the study, the potential impact of the codebase and the systematic study that can inspire future works. Some clarified issues include comments on future research directions and the labeling efficiency plot (which is, however, not analyzed deeper in the main text), and results on additional settings like transfer learning (somewhat preliminary). In the end, two remaining concerns surround whether the technical contribution and the conclusions are sufficiently solid, including

* limited insights: Some reviewers comment that the insights are on the lighter side. The authors identify several issues that may affect the performance of the underlying tasks of active learning, and find that the best setting differs across different active learning strategies. But given that the paper offers at best "best practices of training models on actively-queried labels", it is not clear whether the authors achieve their claimed goal of "compare different strategies in a fair way"---in particular, the conclusion for this particular comparison seems to be missing (e.g. which is recommended in practice, BADGE or LL4AL or others?). Also, given that only three data sets (5 after rebuttal) have been studied in this work (see item below), the "generalization ability" of the conclusions in this paper cannot be clearly established. While the authors provided some additional pieces in the rebuttal, the pieces can use more study to be fully conclusive. Some reviewers are also concerned that the conclusions are rather scattered.

From a practical perspective, it appears to be a chicken-egg problem on whether to fix the active strategy first (and then train the model with the best setting/practice), or fix the training setting first (and then select the best strategy). The authors may want to add more arguments on why they focus on the former rather than the latter.

* limited experiments: several reviewers point out that the few data sets used could not fully justify the "best practice", and demand data sets like ImageNet. The authors offered some new results on TinyImageNet and CIFAR100, but those are not studied as deeply as other data sets at the current point. A more careful study on the two (and other) data sets are thus strongly recommended.